# A multi-agent reinforcement learning framework for exploring dominant strategies in iterated and evolutionary games

Qi Su [1,2,3,7] ✉, Hongyu Wang [4,7], Yu Xia [1] & Long Wang [4,5,6] ✉

Exploring dominant strategies in iterated games holds theoretical and practical significance across diverse domains. Previous studies, through mathematical analysis of limited cases, have unveiled classic strategies such as tit-for-tat, generous-tit-for-tat, win-stay-lose-shift, and zero-determinant strategies. While these strategies offer valuable insights into human decision-making, they represent only a small subset of possible strategies, constrained by limited mathematical and computational tools available to explore larger strategy spaces. To bridge this gap, we propose an approach using multi-agent reinforcement learning to delve into complex decision-making processes that go beyond human intuition. Our approach has led to the discovery of a strategy that we call memory-two bilateral reciprocity strategy. Memory-two bilateral reciprocity strategy consistently outperforms a wide range of strategies in pairwise interactions while achieving high payoffs. When introduced into an evolving population with diverse strategies, memory-two bilateral reciprocity strategy demonstrates dominance and fosters higher levels of cooperation and social welfare in both homogeneous and heterogeneous structures, as well as across various game types. This high performance is verified by simulations and mathematical analysis. Our work highlights the potential of multi-agent reinforcement learning in uncovering dominant strategies in iterated and evolutionary games.

Cooperation is crucial for the survival and flourishing of human society[1]. However, cooperation, an altruistic behavior involving incurring a cost to benefit others, can reduce one's immediate interests, placing oneself at a disadvantage. Understanding the emergence and maintenance of cooperation has long been an evolutionary puzzle[2–4]. Over the past decades, numerous studies have explored this topic, proposing various mechanisms to explain cooperation, such as network reciprocity[4–9], direct reciprocity[1,2,10], indirect reciprocity[11–13], and kin selection[14,15]. Among these mechanisms, direct reciprocity is

one of the most studied. It considers scenarios where two individuals interact repeatedly, leading them to expect reciprocal actions from their partners in future encounters[16,17].

The most challenging and attractive aspect of repeated interactions is designing strategies to achieve payoff dominance over opponents while guiding the entire population to a high level of cooperation[10]. Studies along this line hold both theoretical and practical significance. Over the past decades, several well-known strategies have emerged, which are simple yet effectively model human decision-

[1]School of Automation and Intelligent Sensing, Shanghai Jiao Tong University, Shanghai, China. [2]Key Laboratory of System Control and Information Processing, Ministry of Education of China, Shanghai, China. [3]Shanghai Key Laboratory of Perception and Control in Industrial Network Systems, Shanghai, China. [4]Center for Systems and Control, College of Engineering, Peking University, Beijing, China. [5]School of Advanced Manufacturing and Robotics, Peking University, Beijing, China. [6]Center for Multi-Agent Research, Institute for Artificial Intelligence, Peking University, Beijing, China. [7]These authors contributed equally: Qi Su, Hongyu Wang. ✉e-mail: qisu@sjtu.edu.cn; longwang@pku.edu.cn

making patterns. For instance, in Axelrod's computer tournament, the "tit-for-tat" strategy (TFT), which starts by cooperating and then mimics the partner's previous move, has consistently outperformed other strategies. However, TFT has shortcomings, notably its vulnerability to exploitation by certain strategies. To address this, researchers proposed "generous-tit-for-tat" (GTFT), which starts with cooperation and occasionally cooperates even after a partner's defection. GTFT enhances robustness by combining reciprocity with occasional generosity[18,19]. Another strategy, "win-stay-lose-shift" (WSLS), involves persisting with the current choice after a successful interaction and switching after an unsuccessful one, emphasizing adaptability and maximizing wins[18,20]. A significant recent development is the discovery of the "zero-determinant" (ZD) strategy, which allows a player to unilaterally manipulate their opponents' payoffs to ensure an advantageous outcome, showcasing its exploitative nature[10,21–24].

Indeed, these classic strategies are insightful and have inspired many subsequent studies that use mathematical analysis. However, they represent only a small fraction of the entire space of repeated strategies. Due to the lack of mathematical and the pursuit of analytical proof, most prior studies in this area have been limited to memory of just one step. For example, in two-player two-action games with one-step memory, each player's strategy can be described by four parameters, corresponding to four action profiles. Increasing memory exponentially increases the dimension of the strategy space, making mathematical results infeasible[25,26]. Human cognitive capacity allows for longer memory, and advancements in artificial intelligence enhance information processing. Therefore, analyzing the existence of more powerful strategies in the vast strategy space is crucial[27].

Several recent studies based on human decision patterns have identified a few heuristic strategies such as Hold-a-Grudge (perpetually defecting once the opponent defects)[28], Fool-Me-Once (issuing a warning and forgiving the opponent once for defection, but defecting indefinitely upon subsequent offenses)[28], Omega-Tit-for-Tat (OmegaTFT, TFT with adaptability and deadlock detection)[29], and Gradual-Tit-for-Tat (GradualTFT, TFT with escalating punishment for defection)[30]. These strategies have proven to perform better in Axelrod's tournaments, deriving higher payoffs than classic strategies like TFT, GTFT, and WSLS in pairwise interactions. However, two questions remain largely unexplored: Can these strategies evolve in a large-scale population over the long term? Can they coordinate the population to achieve higher levels of cooperation, beyond merely outperforming other strategies? Addressing these questions poses a significant challenge, necessitating the development of methodologies capable of uncovering strategies within this largely unexplored territory. This area remains largely uncharted in current research, underscoring the need for fresh approaches and paradigms in the study of strategic interactions.

Recent advances in reinforcement learning have substantially enriched the toolkit for studying strategic behavior in iterated games. Unlike traditional game-theoretic approaches that typically assume fixed payoff matrices and limited memory, reinforcement learning enables agents to adaptively update strategies based on past experiences. This flexibility enables the discovery of strategies in complex environments that would be difficult to identify using traditional approaches. Recent studies have successfully applied reinforcement learning to group-structured and networked settings, uncovering forms of cooperation that often transcend human intuition[31,32]. For instance, Xu et al. developed a Q-learning framework on higher-order networks to study cooperation under dynamic environments, showing that reinforcement learning can effectively regulate agent activation and sustain cooperation within groups[31]. Similarly, Jia et al. introduced a reinforcement learning model distinguishing local and global stimuli, revealing how different feedback mechanisms shape conditional cooperation and learning dynamics in social dilemmas[32]. These studies

highlight the potential of reinforcement learning as both a modeling tool and a theoretical lens to explore the evolution of strategies.

In this study, we propose a multi-agent reinforcement learning approach to explore dominant strategies in iterated and evolutionary games. Here, a dominant strategy refers to a strategy that exhibits an evolutionary advantage over others, either by eventually taking over the population under evolutionary dynamics or by maintaining a high frequency in the evolutionarily stable state. In this approach, each agent's strategy is encoded in a Q-table, which enumerates all possible joint action profiles of interacting individuals and maps them to the expected long-term payoffs for selecting each action in a given profile. Agents are trained by interacting with classic strategies and heuristic strategies, and adaptively adjust the Q-table to maximize two objectives: the relative advantage over opponents and their own payoffs. Our approach is general and enables the exploration of any large strategy space and any long interacting memory. Remarkably, we uncover the memory-two bilateral reciprocity (MTBR) strategy by using our approach. MTBR dynamically adapts in repeated games, consistently achieving higher payoffs against nearly all strategies studied. Furthermore, the introduction of MTBR into a large-scale population increases the global payoff, consistent across various payoff structures and population structures. This discovery highlights the effectiveness of reinforcement learning in revealing dominant strategies amidst the complex landscape of strategic interactions.

## Results

### Overview of the multi-agent reinforcement learning framework
To move beyond heuristic strategy design, we introduce a general multi-agent reinforcement-learning framework for evolutionary games that searches directly in the space of finite-memory strategies (see the "Methods" section for details of the framework and definitions of notations appearing below). Each player is modeled as an adaptive agent that maps limited interaction histories to actions via an $N_{\text{state}} \times M$ Q-table and optimizes long-run performance against a diverse opponent pool. The formulation is agnostic to the game matrix, the action set, and the memory length, providing a principled route to strategy discovery rather than hand-crafting. To furnish informative and stable learning signals, the opponent pool includes mentors implementing canonical benchmark strategies, which serve as fixed references during training. Applying this framework to iterated and evolutionary settings, we uncover the memory-two bilateral reciprocity strategy, which achieves a robust balance between resisting exploitation and rapidly building mutual cooperation. Details are provided in the Methods and Supplementary Information.

### Memory-two bilateral reciprocity strategy
Our approach based on multi-agent reinforcement learning reveals a strategy, which we term memory-two bilateral reciprocity strategy, that is effective in fostering cooperation and achieving high payoffs. By delving into its Q-table, we summarize the main decision-making patterns of MTBR: (i) With only one-step memory (i.e., in the second round of a 20-round game), if the opponent opts for "defect" in the first round while MTBR chooses "cooperate", MTBR continues to choose "cooperate" in the subsequent round. This choice likely aims to enhance the probability of the opponent reciprocating cooperation in future interactions. By forgiving the opponent's initial defection, MTBR may quickly foster a state of mutual cooperation. (ii) If both MTBR and the opponent select "defect" in the two preceding rounds, MTBR switches to "cooperate" in the following round. This behavior likely seeks to break the cycle of mutual defection, encouraging both parties to return to cooperative interactions. (iii) In other scenarios, MTBR adopts the action taken by the opponent in the last round.

We then offer insights into MTBR strategies and elucidate the mechanisms underlying their dominance over a wide range of other strategies. We use examples of partial strategies interacting with

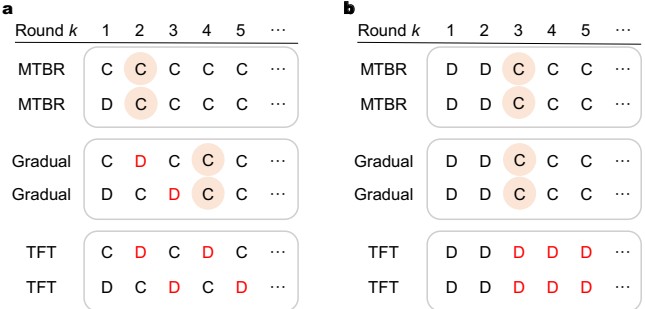

**Fig. 1 | Interactions between MTBRs, GradualTFTs, and TFTs in repeated Prisoner's Dilemma.** The ability to swiftly establish mutual cooperation after initial defection is a key factor in the success of strategies in the repeated Prisoner's Dilemma. Yellow circles represent individuals transitioning from defection to mutual cooperation, while red letters indicate increased defection compared to MTBR, resulting in lower overall payoffs. **a** Two identical individuals randomly select "cooperate" and "defect" in the first round. MTBRs quickly establish mutual cooperation, while TFTs get trapped in a cycle of cooperation and defection. **b** Two identical individuals both defect in the first round. Both MTBRs and GradualTFTs are able to recover to mutual cooperation, while TFTs continue to defect indefinitely against each other.

themselves to illustrate the distinct behavioral patterns of MTBR. In Fig. 1, we depict scenarios where two MTBRs, two GradualTFTs, and two TFTs interact. As depicted in Fig. 1a, when two MTBRs initially choose "cooperate" and "defect" respectively, they quickly establish a cooperative state. In contrast, two GradualTFTs require three rounds to reach cooperation, while two TFTs fall into a perpetual "cooperate-defect" cycle under similar circumstances. In Fig. 1b, when two MTBRs initially defect, they achieve cooperation after two rounds. The behavior of GradualTFTs in this scenario mirrors that of MTBR, while TFTs continue in a cycle of mutual defection. It is noteworthy that in a noiseless repeated Prisoner's Dilemma, initial mutual cooperation often leads to sustained cooperative interactions across most strategies. Furthermore, we demonstrate the effectiveness of MTBR when interacting with other strategies, such as TFT and GradualTFT (see Supplementary Fig. 1).

## MTBR strategy effectively drives the entire population toward a more prosperous state

Our approach enables the exploration of games of arbitrary length, with arbitrarily large action sets, and strategies of arbitrary memory. In the main text, we focus on the 20-round Prisoner's Dilemma games with two-step memory strategies. After training, we evaluate the performance of the obtained strategies using classical parameters: $R = 3$, $S = 0$, $T = 5$, and $P = 1$. This setup creates a robust dilemma in which defection yields a higher payoff, making it challenging for cooperative behaviors or traits to survive and evolve. Additionally, we explore a broad range of interaction lengths and game types to evaluate the performance and adaptability of the MTBR under various conditions.

In this section, we examine an interacting system composed of various strategies and investigate how the introduction of MTBR affects the outcomes of these interactions. We begin by exploring a group of seven well-recognized strategies in the repeated Prisoner's Dilemma, collectively referred to as set 1, which includes GradualTFT, OmegaTFT, TFT, GTFT0.3, Fool-Me-Once, WSLS, and Hold-a-Grudge (see Fig. 2a). The purple bars in Fig. 2a represent the average payoff of each strategy. Intriguingly, the addition of MTBRs to the population leads to a significant increase in the average payoff for each strategy within the group, as well as for the population as a whole. Furthermore, MTBR emerges as the second-highest performing strategy, closely behind GradualTFT. GradualTFT proves to be MTBR's most challenging opponent in these repeated games. The difference in payoff

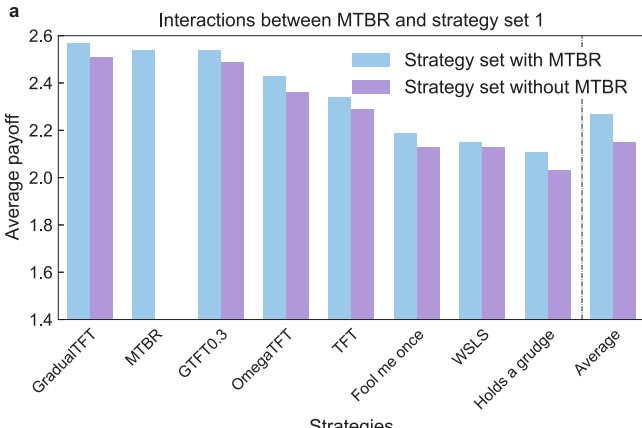

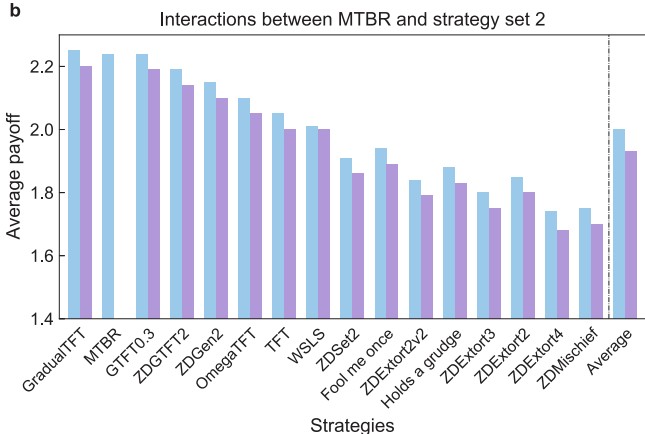

**Fig. 2 | MTBR effectively drives the entire population towards a state of high payoffs.** We consider two sets of strategies: set 1 (comprised of GradualTFT, OmegaTFT, TFT, GTFT0.3, Fool-Me-Once, WSLS, and Hold-a-Grudge) and set 2 (comprised of eight zero-determinant strategies besides those in set 1). **a** The interacting outcome in a population comprised of solely strategy set 1 (purple bars) and the strategy set 1 with MTBR (blue bars), namely, the average payoff of each strategy when interacting with every other strategy in the set. **b** The interacting outcome in a population comprised of only strategy set 2 (purple bars) and the strategy set 2 with MTBR (blue bars). The introduction of MTBR increases the average payoff of the whole population. MTBR achieves higher average payoffs in both scenarios. Each interaction consists of a 20-round repeated Prisoner's Dilemma, and the results are averaged over 10,000 repeated experiments.

between MTBR and GradualTFT is remarkably small. This observation supports the idea that MTBR possesses exceptional properties that promote cooperation within the population. It not only maintains high payoffs for itself but also drives the entire population toward a more prosperous state (i.e., enhance the average payoffs across the population), exerting a positive social influence. Conversely, strategies characterized by exploitation tend to exert a negative social influence, diminishing the overall benefits to the population.

In addition to the classic strategies mentioned earlier, we expand our exploration to include eight representative zero-determinant strategies, which we refer to as set 2 (see "Methods" section for details). These strategies, known for their ability to unilaterally control opponents' payoffs, introduce complexity to the population dynamics. Figure 2b demonstrates that in the absence of MTBR strategies, the introduction of zero-determinant strategies decreases the average payoffs from 2.15 to 1.93 (as seen in the purple bars in Fig. 2a and b). This decrease is evident across all strategies, indicating the negative influence of zero-determinant strategies on the overall population performance. Once again, we observe a significant increase in the average payoffs of the population with the introduction of MTBR strategies (see Supplementary Fig. 2 for the interacting outcome for

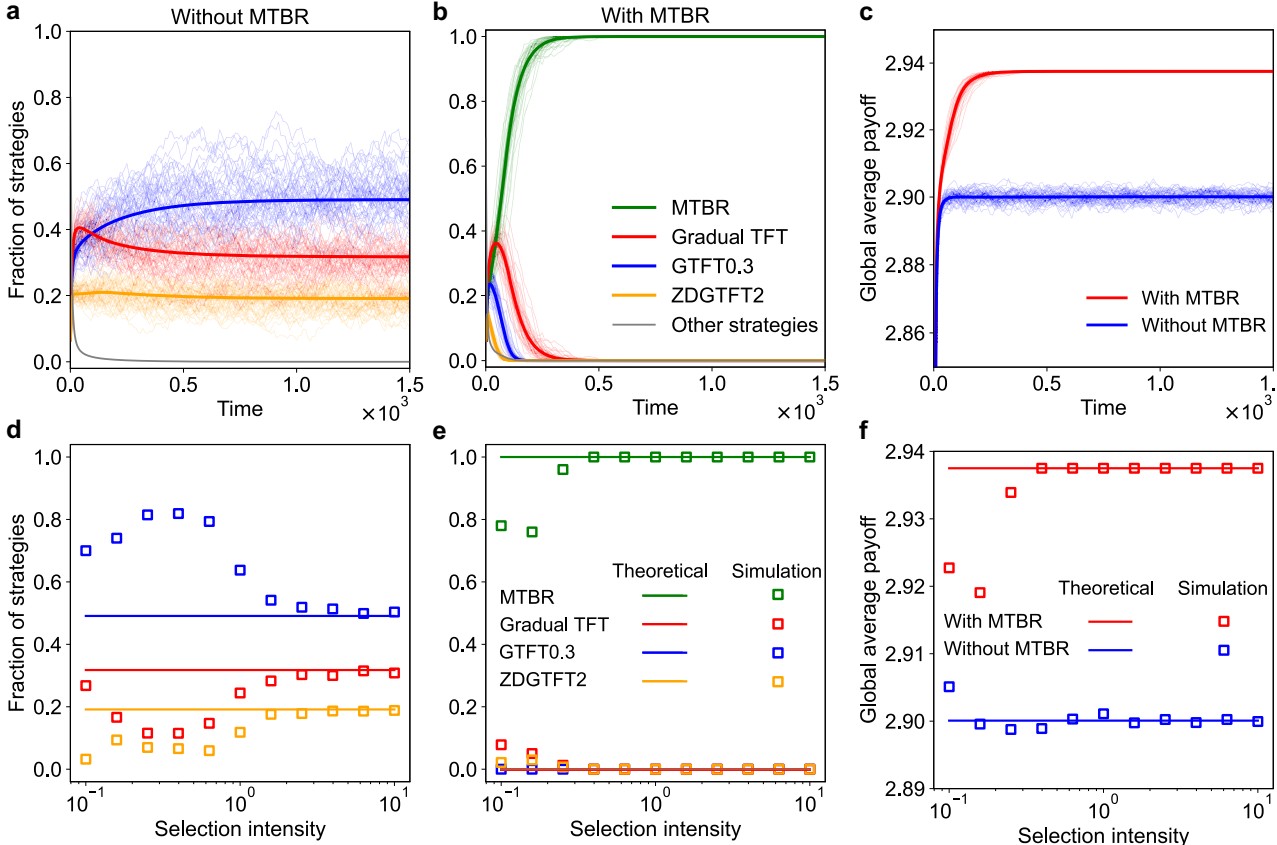

**Fig. 3 | Memory-two bilateral reciprocity strategy dominates and drives the population to a more prosperous state in an evolutionary process.** We consider two evolving populations, one comprised of the strategy set 2 (see "Methods" section for details) without MTBR and one comprised of the strategy set 2 with MTBR. **a** Evolutionary trajectories of an evolving population comprised of the strategy set 2 without MTBR. **b** Evolutionary trajectories of an evolving population comprised of the strategy set 2 with MTBR. **c** The average payoff of the population throughout the evolutionary process, with blue and red curves representing scenarios without and with MTBR, respectively. The solid lines represent the evolutionary trajectories predicted by the replicator equations, while the dashed lines depict the results of 50 repeated experiments. **d** Impact of selection intensity on the equilibrium frequency of strategies without MTBR. **e** Impact of selection intensity on the equilibrium frequency of strategies with MTBR. **f** Impact of selection intensity on the equilibrium average payoff with (red) and without (blue) MTBR. Parameters: selection intensity $\delta = 3.0$ (**a**, **b** and **c**), population size $N = 7000$ without MTBR and $N = 7500$ with MTBR.

each pair of strategies). Furthermore, MTBR strategies achieve much higher payoffs than nearly all other strategies. This underscores their effectiveness in promoting cooperation within the population, leading to elevated collective payoffs.

## The dominance of MTBR in evolving populations

So far, we have considered interactions within a population comprised of MTBR and a set of representative strategies, where individuals' strategies are fixed throughout the evolutionary process. In this part, we allow for the evolution of individuals' strategies, meaning strategies yielding a higher payoff are likely to be imitated and used by more individuals. We consider an evolving population where initially each individual randomly chooses one of the strategies between MTBR and strategy set 2. Then, in each generation, all individuals are paired and play games for twenty rounds to derive an average payoff, using the same payoff structure as above, $R = 3$, $S = 0$, $T = 5$, $P = 1$. After all interactions, a random player $i$ is selected to update his strategy, and another random player $j$ is selected. Player $i$ imitates player $j$'s strategy with probability

$$p_{i \to j} = \frac{1}{1 + \exp\left[\delta(\overline{U}_i - \overline{U}_j)\right]}. \tag{1}$$

Here, $\delta \in [0, \infty)$ is the selection intensity parameter. It determines how strongly the payoff difference influences the probability of

strategy adoption. When $\delta$ is close to zero, the selection is nearly random, while larger values of $\delta$ make the selection more dependent on the payoff difference.

We first consider an evolving population without MTBR. Figure 3a shows that GTFT0.3 becomes the most abundant strategy, followed by GradualTFT and ZDGTFT2, with the other strategies becoming almost extinct. When the population evolves to a state comprised of these three strategies, cooperation becomes frequent, leading to an average payoff increase to 2.900, a remarkably high level (see blue lines in Fig. 3c). Next, we examine a scenario in which MTBR is introduced into the population, as illustrated in Fig. 3b. Remarkably, MTBR emerges as a dominant force in the evolutionary process. Initially, the frequencies of MTBR, GradualTFT, and GTFT0.3 all increase. However, as the population stabilizes, GTFT0.3 quickly declines and eventually becomes extinct. Over a longer period, GradualTFT also declines and ultimately disappears from the population. One possible explanation for this outcome is that both MTBR and GradualTFT achieve high payoffs in the early stages of the evolutionary game (see red and green lines in Fig. 3b). As evolution progresses, exploitative strategies gradually decrease in frequency, with MTBR and GradualTFT becoming the predominant strategies in the population. While MTBR and GradualTFT yield the same payoff when they interact with each other, MTBR benefits from a higher payoff in self-play compared to GradualTFT. This advantage allows MTBR to increasingly dominate the population over time. Furthermore, we observe that the introduction

of MTBR increases the average payoff of the entire population. Specifically, the average payoff across the population rises to 2.938 with the inclusion of MTBR (see red lines in Fig. 3c). Interactions involving MTBR yield higher payoffs compared to those involving GradualTFT, explaining this increase.

While this is only a 0.038 increase, it is noteworthy given the narrow margin between 2.900 and the theoretical maximum of 3.000. Considering the stochastic nature of initial interactions, this gain, amounting to 38% of the remaining possible improvement, is substantial. Besides, the cooperation-promoting effect of MTBR becomes even more pronounced in settings where GTFT does not evolve. For instance, in a small population of ~50 individuals (Supplementary Fig. 3), the absence of GTFT0.3 results in a low average payoff of 1.78. Introducing GTFT0.3 increases the average payoff to 2.44, indicating partial cooperation. However, introducing MTBR instead leads to a much higher average payoff of 2.94, nearly reaching the full cooperation payoff. This demonstrates that MTBR can outperform GTFT0.3 even in more challenging evolutionary settings. Furthermore, when both MTBR and GTFT0.3 are introduced, the resulting dynamics again yield a high average payoff, underscoring MTBR's robust cooperation-promoting capabilities.

Furthermore, we analyze the evolutionary dynamics across a wide range of selection strengths, from weak selection (i.e., $\delta = 0.1$) to strong selection (i.e., $\delta = 10$). In nearly all cases, MTBR consistently dominates over all other strategies across this spectrum of selection strengths (see Fig. 3e). The introduction of MTBR also drives the entire population toward a more prosperous state for any selection strength (see Fig. 3f). Additionally, we present mathematical results regarding the evolutionary trajectories and outcomes (see "Methods"). In most instances, these mathematical results accurately predict the evolutionary direction observed in simulations (see Fig. 3).

In addition to noise-free conditions, we also examine the evolutionary dynamics under behavioral noise. While our model focuses on deterministic strategy execution, prior studies have designed strategies that are explicitly robust to noise. Yi et al.[33] proposed TFT-ATFT, a memory-two strategy which can efficiently recover from implementation errors without being exploited. Murase and Baek[34,35] identified a class of "friendly rival" strategies that combine partner-like cooperation with rival-like defensibility; their evolutionary robustness emerges under implementation error and is further strengthened in structured populations with long memory. In our model, we incorporate strategy execution noise, wherein agents occasionally mis-implement their intended actions. This form of behavioral uncertainty has become a standard approach to modeling imperfect decision-making in repeated games[18,36,37]. We explore three levels of execution noise: $\eta = 0.001$, $0.01$, and $0.05$. Our results reveal that MTBR remains robustly dominant under low noise levels (see Supplementary Fig. 4). Even at $\eta = 0.01$, MTBR continues to outperform all other strategies by a wide margin. When the noise level is further raised to $\eta = 0.05$, MTBR and GTFT0.3 jointly dominate the evolutionary dynamics, consistently outperforming all other strategies. These findings underscore the robustness of MTBR's evolutionary advantage in noisy environments.

### The dominance of MTBR in various games and networks

Beyond the specific parameters of the Prisoner's Dilemma, where defection yields a larger payoff than cooperation in one-shot games, our research now extends to a broader range of games, including Snowdrift Games (SG, where the optimal strategy is to act opposite to the opponent), Stag-hunt Games (SH, where the equilibrium is either mutual cooperation or mutual defection), and Harmony Games (HG, where the equilibrium is mutual cooperation). We examine four types of games, characterized by two parameters: greediness $T - R$ and unfear $S - P$. Variations in these parameters reflect different levels of dilemma strength, as described in prior work[38].

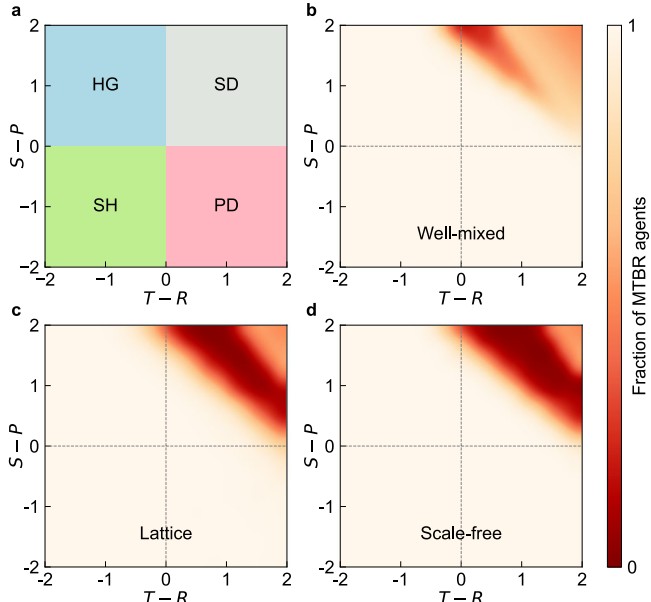

**Fig. 4 | MTBR plays a dominant role in various games and networks.** We explore four different types of games, measured by greediness $T - R$ and unfear $S - P$. **a** The value of $T - R$ and $S - P$ lead to the Prisoner's Dilemma (PD, the lower-right quadrant) for $T - R > 0$ and $S - P < 0$, Snowdrift Games (SG, the upper-right quadrant) for $T - R > 0$ and $S - P > 0$, Stag-hunt Game (SH, the lower-left quadrant) for $T - R < 0$ and $S - P < 0$, Harmony Game (HG, the upper-right quadrant) for $T - R < 0$ and $S - P > 0$. **b** Fraction of MTBR strategies in the completely connected networks after a long-term evolutionary process. **c** Fraction of MTBR strategies in the lattice networks. **d** Fraction of MTBR strategies in the scale-free networks. Parameters: $R = 3$, $P = 1$, $\delta = 1$, $N = 10,000$.

As shown in Fig. 4, MTBR demonstrates dominance across nearly all game types, underscoring the effectiveness of the multi-agent learning approach in developing robust strategies. Nonetheless, in Snowdrift Games, particularly along the line where $2R = T + S$, MTBR's prevalence declines. In this region, GTFT0.3 becomes the dominant strategy, with a smaller proportion of GradualTFT individuals also present in the population. In the upper-right section of the payoff matrix, where both $T$ and $S$ increase (with $2R < T + S$), the population gradually shifts toward a mixed state dominated by MTBR and TFT. The closer the population is to this upper-right corner, the higher the proportion of TFT individuals. This trend indicates that MTBR struggles specifically along the $2R = T + S$ line. Near this boundary, GTFT0.3 and GradualTFT gain an advantage, but as $T$ and $S$ increase, MTBR's frequency first rises and then declines, with GTFT0.3 and GradualTFT quickly vanishing. Eventually, TFT becomes increasingly predominant in the evolutionary stable population. Optimizing strategies specifically for the Snowdrift Games may be achieved by using the Snowdrift Games payoff matrix as a training condition.

This phenomenon can be understood by examining the inherent dynamics of Snowdrift Games, where TFT's success is more prominent under specific payoff conditions. Specifically, in regions closer to the upper-right corner of the payoff matrix, where $T + S > 2R$, the incentives in Snowdrift Games align such that alternating between cooperation and defection yields a higher payoff than mutual cooperation. This asymmetry in rewards reduces the advantage of cooperation-promoting strategies like MTBR in these scenarios. The nature of Snowdrift Games favors strategies that can balance cooperation with defection, as overly cooperative strategies may be exploited. TFT is better suited to this environment. When $T + S > 2R$, TFT can achieve a higher payoff through the alternation between cooperation and defection, providing it an advantage over MTBR, which predominantly

promotes cooperation. In the evolutionarily stable population along the line $2R = T + S$, GTFT0.3 and GradualTFT emerge as the primary strategies due to their capacity for conditional cooperation balanced with occasional defections. As $T$ and $S$ increase further, MTBR initially becomes more competitive, but as $T$ and $S$ continue to increase, TFT gradually dominates. This trend reflects TFT's payoff benefit derived from its alternation pattern, which is optimal in Snowdrift Games with strong incentives for exploitation. These findings underscore that MTBR's optimal strategic advantage occurs when $2R > T + S$. This relationship is visually represented in Fig. 4, which delineates the boundary condition $2R = T + S$ as critical for MTBR's potential to dominate the population. Further insights are provided in Supplementary Figs. 5 and 6, which detail the cooperation rates and average payoffs in stable states across different network structures and game settings, elucidating the strategic boundaries that influence the emergence and stability of cooperative behaviors within populations.

Recent studies have drawn an important distinction between R-reciprocity and ST-reciprocity in repeated interactions[39,40]. R-reciprocity emphasizes stable mutual cooperation, typically sustained by memory-one or memory-two strategies in the repeated Prisoner's Dilemma. In contrast, ST-reciprocity involves alternating cooperation and defection, which can yield higher payoffs in games like the Snowdrift or Chicken game, especially when the condition $S + T > 2R$ holds. The MTBR strategy developed in our study was trained within a repeated Prisoner's Dilemma setting and relies solely on the history of mutual actions, without access to payoff information. As such, it is well-suited for fostering R-reciprocity but not equipped to recognize or capitalize on payoff structures that favor ST-reciprocity, offering a plausible explanation for its reduced or even vanishing dominance for payoff structure satisfying $S + T > 2R$. These observations suggest that supporting ST-reciprocity may require agents to incorporate payoff-based representations or be trained in environments explicitly designed to reward such alternation. We anticipate that future research will explore this promising direction.

Besides the well-mixed population, we investigate MTBR's adaptability in more complex network structures. Recent studies have highlighted the influence of complex population structure on the emergence and diffusion of cooperation, emphasizing the significance of interindividual connectivity[9,41–47]. This study evaluates MTBR's performance in complex network structures, gaining insights into its robustness and adaptability across different spatial architectures. We examine three different spatial structures: well-mixed population, lattice grid networks without periodicity, and scale-free networks with an average degree of 4. Scale-free networks are generated using the Barabási-Albert algorithm[48]. Figure 4b–d shows the proportion of MTBR in evolutionarily stable states for well-mixed populations, lattice grid networks, and scale-free networks, respectively. These results are based on analyses from over 1000 simulations under different game payoff matrices. We first focus on the bottom-right parts of Fig. 4b–d, where the social dilemma type is Prisoner's Dilemma. MTBR exhibits remarkable dominance in all three spatial structures. Additionally, we observe in experiments that in lattice grid networks and scale-free networks, the speed at which MTBR occupies the entire population is significantly slower than in the well-mixed population. This is due to the relatively poor connectivity in lattice and scale-free networks, especially for boundary nodes in lattice and small nodes in scale-free networks. The slower spread of strategies is a consequence of the limited connectivity of the networks, affecting the speed at which MTBR establishes dominance in the population.

## Extensions

In this section, we explore several extensions of the multi-agent reinforcement learning framework. During training, agents are stochastically matched with either mentors or other agents. In the default setting, we use an equal number of agents and mentors, and each agent has an equal probability of being matched with a mentor or another agent. We have also examined alternative configurations, for instance, assigning each agent a 25% probability of interacting with a mentor (and 75% with another agent), or vice versa. These variations in the mentor encounter probability lead to different training dynamics. In particular, when agents are more likely to interact with mentors, MTBR tends to emerge more frequently across training runs. Notably, even when the probability of encountering a mentor is relatively low, MTBR still emerges in some cases. This indicates that the multi-agent reinforcement learning approach can still significantly reduce the effective strategy search space compared to unguided exploration, thereby facilitating the discovery of functional strategies under diverse conditions. Furthermore, while the probability of mentor interaction may influence the specific strategies that emerge, in our studies, it does not affect the overall convergence of the reinforcement learning process.

To assess the evolutionary robustness of MTBR, we incorporated strategy mutation into the evolutionary process. At each update step, with probability $\mu$, individuals adopt a randomly selected strategy from a predefined pool rather than imitating another individual[49]. We investigated three representative mutation rates: $\mu = 0.001$, $\mu = 0.01$, and $\mu = 0.1$. At the lowest rate ($\mu = 0.001$), the evolutionary dynamics closely mirror those of the mutation-free scenario: MTBR quickly dominates the population, with GradualTFT emerging as the nearest competitor. At $\mu = 0.01$, MTBR continues to dominate reliably, although the fixation process is marginally slower. When the mutation rate increases to $\mu = 0.1$, the population no longer converges to a monomorphic state. Instead, MTBR retains a substantial majority (~77%), while GradualTFT and GTFT0.3 persist at stable levels of 9% and 2%, respectively (see Supplementary Fig. 7). These results demonstrate that MTBR remains the most abundant and evolutionarily dominant strategy across a wide range of mutation rates, highlighting its exceptional robustness to stochastic perturbations.

The impact of memory length on strategy dynamics has attracted considerable attention in evolutionary game theory[20,21,50–54]. Recent advances have proposed several longer-memory strategies that exemplify sophisticated mechanisms for sustaining cooperation. For instance, Li et al.[52] introduced the CURE strategy, which embodies a cumulative reciprocity mechanism that enables players to release strategic goodwill in a controlled and persistent manner. Hilbe et al.[53] developed the AON2 strategy based on an axiomatic framework, offering a rare example of provably stable cooperation in memory-$n$ strategy spaces. Glynatsi et al.[54] further demonstrated that extending memory length allows for conditional cooperation schemes that remain robust in complex stochastic settings. These works have significantly advanced our understanding of how memory-depth can be leveraged to promote strategic stability and resilience. To situate our findings within this broader literature, we included CURE, AON2, and Reactive-2-Partner in our evolutionary simulations. MTBR maintains its dominance even under such competitive conditions (see Supplementary Fig. 8). Specifically, it fully takes over the population under $b/c = 2$, and stabilizes at ~98% under $b/c = 1.5$, reflecting its robustness against a diverse and sophisticated strategic environment.

A central challenge in the study of repeated social dilemmas is to identify strategy classes that can sustain long-term cooperation while remaining resistant to exploitation. Hilbe et al. introduced the concept of partner strategies, defined by two key criteria[10]: (i) the strategy achieves mutual cooperation when interacting with itself, and (ii) no other strategy can obtain a higher payoff against it than the mutual cooperation reward $R$. These strategies have garnered considerable attention for their ability to stabilize cooperation in infinitely repeated games without requiring payoff discounting. Stewart and Plotkin[55] demonstrated that partner strategies, as defined by Hilbe et al., exhibit evolutionary robustness under a broad range of conditions. Extending this line of work, we provide a formal proof that our proposed strategy, MTBR, satisfies both conditions required by the definition of a partner

strategy across a wide range of parameters. In addition, we show that MTBR is evolutionarily robust, meaning it cannot be selectively replaced by mutant strategies in large populations. Detailed mathematical derivations and algorithmic verification are provided in Supplementary Note 3. Together, these findings establish MTBR as both a high-performing strategy in simulations and a theoretically grounded partner strategy with strong evolutionary guarantees.

## Discussion

Humans are inherently social beings, constantly engaging in interactions within society for survival and daily life. One of the fundamental challenges has been to discover effective strategies that maximize the benefits of these interactions. This quest has a long history, dating back to the origins of living systems. Recent studies exploring dominant strategies in repeated games have advanced research and development in this field. However, many of these studies have relied to some extent on intuition[56–58]. While this approach has led to the discovery of well-known strategies such as tit-for-tat and win-stay-lose-shift, it falls short when exploring a large strategy space, particularly strategies with long memory[51,59–61]. To address this limitation, we have developed a multi-agent learning approach. This methodology enables the exploration of arbitrarily complex strategies, providing a more comprehensive understanding of strategic interactions in social settings.

Our study, grounded in multi-agent reinforcement learning, uncovers a strategy termed memory-two bilateral reciprocity strategy, which consistently outperforms nearly all previously known strategies. Following established conventions in the literature, we define "outperform" to mean that MTBR achieves a higher average payoff when interacting with a diverse population of strategies, rather than maximizing payoff in every individual encounter[61,62]. We attribute MTBR's strong performance primarily to its particular approach to forgiveness. In the second round of repeated games, MTBR chooses to forgive the opponent's initial random defection when MTBR itself initially cooperated. This choice appears strategic, anticipating that mutual defection in the first round likely leads to opponent retaliation, resulting in significant losses for MTBR if it chooses not to forgive. However, if MTBR cooperates in the first round, the opponent is more likely to cooperate in the second round, resulting in a mutually beneficial outcome if MTBR also cooperates in the second round. During the game, if both MTBR and the opponent defect in the previous two rounds, MTBR chooses to cooperate once to attempt to break the deadlock. This decision is based on the understanding that in most game settings, especially in the Prisoner's Dilemma, mutual defection leads to very low payoffs. Therefore, it is beneficial for MTBR to attempt to break the cycle of defection to increase its own payoff.

It is well recognized that most strategies including widely studied ones such as win-stay lose-shift and generous tit-for-tat are vulnerable to exploitation by the unconditionally defecting strategy AllD in direct pairwise interactions, due to AllD's refusal to cooperate. However, the central goal in this domain is to identify strategies that not only secure high average payoffs across heterogeneous opponents but also demonstrate robustness to invasion in evolving populations. In addition to outperforming all other strategies, MTBR demonstrates a competitive edge in evolving populations, where individuals tend to forgo their current strategy and switch to those bringing higher payoffs. We have rigorously evaluated the generality and robustness of MTBR by testing it against a broad and representative set of strategies. This set includes classical strategies (TFT, GTFT0.3, WSLS), AllD-like strategies (Fool-Me-Once, Hold-a-Grudge), manually crafted complex strategies (OmegaTFT, GradualTFT), eight Zero-Determinant (ZD) strategies, several recently proposed strategies that demonstrate high performance in evolving populations, such as CURE, AON2, and Reactive-2-Partner (see Supplementary Fig. 8), and over 10,000

memory-one strategies uniformly sampled from the full memory-one strategy space (see Supplementary Fig. 9). Across this diverse landscape, MTBR consistently demonstrates superior evolutionary performance.

Furthermore, we explored the performance of MTBR under various population structures and payoff matrices. Our results indicate that MTBR consistently performs well across diverse settings, including well-mixed populations, lattice networks, and scale-free networks, showing no significant variation in its performance across these different network structures. We validated that the selection intensity has minimal impact on the probability of MTBR dominating the population, only affecting the speed at which the population reaches evolutionarily stable states (see Supplementary Fig. 10a). Additionally, we verified that MTBR's dominance persists for populations with $N > 150$, which is common in evolutionary game simulations (see Supplementary Fig. 10b). Although the discovery of MTBR is based on a setup of twenty rounds, we have also examined other interaction lengths to ensure that our findings are not limited to this particular setting. When the horizon is extremely short, even highly effective cooperative strategies cannot calibrate their behavior or establish sustained reciprocity, and thus many strategies, including MTBR, are unable to succeed. By contrast, when the interaction length becomes very long (or in the limit of infinitely repeated games), the self-play payoff of cooperative strategies converges to $R$, thereby reducing performance differences among them. Nevertheless, extending the horizon to 1,000 rounds under noise-free conditions shows that MTBR remains highly successful: in about 90% of independent runs it ultimately takes over the entire population, while GTFT0.3 dominates in the remaining 10% of runs (see Supplementary Fig. 11). This result confirms that MTBR's cooperative advantage is not restricted to the 20-round setting, although its relative advantage becomes proportionally diluted as the horizon increases.

In addition to its advantages over all other strategies, MTBR demonstrates the ability to coordinate the population to reach a significantly higher payoff than when MTBR is absent. In evolutionary game theory, successful strategies often exhibit the ability to balance cooperation and competition, adapt to various environments, and tolerate strategy diversity. MTBR exemplifies these characteristics, showing remarkable cooperation with strategies like GradualTFT while effectively defending against exploitative opponents who take advantage of cooperative behavior. Furthermore, MTBR's evolutionary fitness extends beyond outcompeting the simplest one-step memory strategies like TFT or WSLS. Unlike these traditional strategies, which are susceptible to exploitation and lack mechanisms to foster cooperation, MTBR's sophisticated two-step memory feature enables it to withstand exploitation while promoting cooperative behavior. Consequently, in environments with high strategy diversity, where traditional strategies may falter, MTBR emerges as a resilient and competitive contender, showcasing its superiority in evolutionary dynamics.

Cooperation is a central theme in the study of strategic interactions, and understanding its origins has attracted increasing attention across behavioral sciences. A recent study by Shen et al.[63] challenges the long-standing debate between prosocial preference and confused learner hypotheses, showing that human cooperation is shaped by a mixture of motives, including fairness, self-interest, and sensitivity to perceived intentions. Our work takes a complementary perspective. Rather than assuming innate prosocial tendencies, we employ reinforcement learning to explore whether cooperative behavior can emerge from experience-driven adaptation. Interestingly, the strategy that arises (MTBR) exhibits a form of contingent reciprocity: it fosters mutual cooperation with cooperative partners, while resisting exploitation. This echoes the behavioral characteristics of classical strategies like TFT or GradualTFT, which are not designed to exploit others but to

maintain fairness and avoid exploitation. While MTBR is not explicitly designed to be prosocial, it often behaves in a prosocial-like manner by promoting high collective payoff through mutual cooperation. Our results suggest that such behavior can arise naturally from the structure of interaction and the learning process itself, rather than being imposed through fixed preferences. We anticipate that this observation will inspire further research into the underlying mechanisms driving the evolution of prosocial behavior.

MTBR strikes an excellent balance between the amount of information utilized and the clarity of decision-making. This research paves the way for investigating more complex strategies, and we expect substantial exploration in this domain. Despite MTBR's strong performance described above, its effectiveness diminishes as execution errors increase, such as frequently mistakenly cooperating when intending to defect, or vice versa. It is worth noting that rising noise poses significant challenges not only for MTBR but also for other cooperation-oriented strategies. Exploring strategies that remain effective in highly noisy environments is therefore of practical relevance. Further investigation into how memory length impacts strategy effectiveness could yield insights for optimizing memory parameters to enhance performance. Additionally, examining the influence of game matrices during training on agent performance may lead to specialized strategies tailored to specific contexts. Exploring the significance of "first impressions" in strategic interactions and designing more complex strategies based on initial actions could result in approaches with greater adaptive capacity. Expanding research to dominant strategies in multiplayer games, such as public goods games, presents an exciting avenue for future studies. Addressing these aspects not only deepens our understanding of strategic dynamics but also creates opportunities for developing more robust and adaptive strategies in complex social settings.

## Methods
### Multi-agent reinforcement learning
We employ a multi-agent Q-learning approach tailored to evolutionary repeated games. Prior work on multi-agent learning in Markov or stochastic games has established how agents can optimize actions under various interaction structures[64–68]. In contrast to settings that emphasize equilibrium tracking or belief updating, our focus is the repeated Prisoner's Dilemma (PD), a general-sum environment in which agents must balance short-term gains against the long-term benefits of cooperation. We study how learning rules operating on limited memory can give rise to effective strategies over many interactions.

We consider an iterated two-player, two-action matrix game that extends over many rounds. The game involves $N_a = 49$ learning agents and $N_m = 49$ mentors with predefined strategies. In each round, a player can choose either a cooperative or a defective action. Mutual cooperation rewards each player with a payoff $R$, while mutual defection results in a punishment $P$. Unilateral cooperation, where one player cooperates while the other defects, gives the cooperator the sucker's payoff $S$ and the defector the temptation payoff $T$. The relationship between these payoffs $T > R > P > S$ characterizes the well-known Prisoner's Dilemma (PD). The classic single-shot PD offers insights into the immediate incentives for cooperation and defection, but it falls short in explaining the emergence of long-term cooperative behavior. To overcome this limitation, researchers have extended the PD into repeated games, where the same players engage in the PD multiple times. This extension, known as the repeated Prisoner's Dilemma, allows players to adapt their actions based on the history of interactions, providing a valuable approach for studying the evolution of cooperation over time.

Next we introduce a general framework for two players engaged in $L$-round games with $M$ action options and an $\ell$-step memory, where $L, M, \ell$ can be arbitrarily large. For two interacting individuals using strategies with $\ell$-step memory, the total number of possible states, $N_{\text{state}}$, is $N_{state} = (\frac{M^{2\ell+2}-1}{M^2-1} - 1)$ (see Supplementary Note 1). Here, the state refers to the interaction history that an agent uses to make decisions. For an agent employing a memory-two strategy, decisions are based solely on the actions taken by both the agent and its opponent in the two preceding rounds. Specifically, the relevant information includes the opponent's action two rounds ago, the agent's action two rounds ago, the opponent's action in the previous round, and the agent's action in the previous round. We represent the state as a four-element tuple: (opponent's action two rounds ago, agent's action two rounds ago, opponent's action last round, agent's action last round).

Note that the formula of $N_{\text{state}}$ accounts for the additional state types generated in the initial rounds when agents cannot yet accumulate a full $\ell$-length memory due to limited interaction history. Let $\mathbf{s} = \{s_1, s_2, \cdots, s_{N_{state}}\}$ represent the set of all possible states, and $s_{i,t}$ the state observed by agent $i$ at time step $t$, corresponding to the last $\ell$ action pairs of both players. Every agent is assigned an independent $N_{\text{state}}$-by-$M$ Q-table, where the entry in the $i_{\text{th}}$ row and the $j_{\text{th}}$ column reflects the agent's expected long-term cumulative reward for choosing to action $j$ in state $s_i$ (see Fig. 5). When interacting with others, agents determines their actions based on their Q-table, taking into account the current state $s_i$ derived from the last $\ell$ steps of interactions. Additionally, agents explore their action space using an $\epsilon$-greedy strategy: with a probability of $1 - \epsilon$, the agent exploits its current knowledge by selecting the action with the highest Q-value, and with a probability of $\epsilon$, the agent explores by choosing a random action.

We define $I_{\text{total}}$ as the total number of training iterations. In each iteration, we randomly select player $p_1$ from the set of $N_a$ agents and player $p_2$ from the set of $N_a$ agents and $N_m$ mentors. The two selected players then engage in an $L$-round game. Note that in the first round of each repeated game, all players, including both learning agents and mentors, randomly choose their initial action between cooperation and defection (We have also explored alternative initialization schemes, such as always starting with cooperation or defection. However, randomizing the initial action leads to a more robust and meaningful evaluation of strategic responses during training). After completing an $L$-round game, each agent updates its Q-table based on the interaction history, which includes the payoffs received in each round and the overall outcome (win or loss). Specifically, for each agent $X \in \{1, 2\}$, the Q-value corresponding to state $s_{p_X,t}$ and action $a_{p_X,t}$ in each round $t$ is adjusted. Let $U_{p_X,t}$ denote player $p_X$'s payoff in round $t$, with the average payoff over all rounds given by

$$\overline{U}_{p_X} = \frac{1}{L}\sum_{t=1}^{L} U_{p_X,t}. \qquad (2)$$

In our reinforcement learning framework, agents optimize a utility function $W_{p_X}$ that integrates two key components: the agent's relative advantage over its opponent and its own expected payoff. The relative advantage promotes strategies that can consistently outperform others in direct encounters, making them resilient to exploitation. At the same time, achieving high absolute payoffs is crucial for fostering mutual cooperation and coordination, qualities that enhance a strategy's evolutionary success within the broader population. This dual-objective design facilitates the emergence of strategies that are both competitively robust and socially beneficial.

To balance the focus between defeating the opponent and maximizing rewards, we introduce the parameter $\theta$. When $\theta = 0$, the agent focuses solely on defeating the opponent, potentially leading to a strategy of full defection in the Prisoner's Dilemma. Conversely, when $\theta = 1$, the agent focuses on maximizing payoffs, which may increase the likelihood of exploiting the opponent. We assign a reward

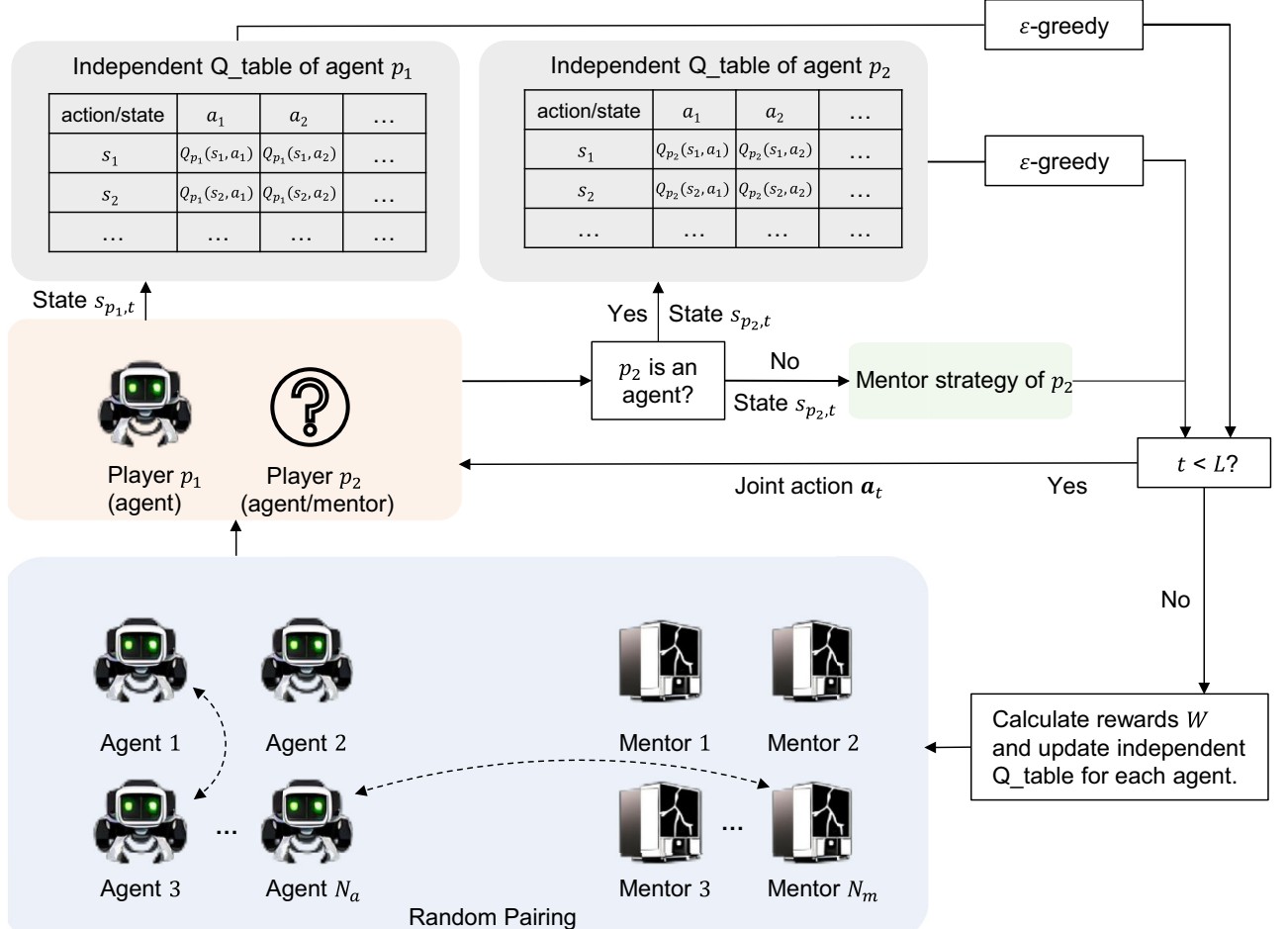

**Fig. 5 | Illustration of multi-agent reinforcement learning for exploring dominant strategies in iterated games.** We consider a population comprised of $N_a = 49$ agents and $N_m = 49$ mentors. Each agent is equipped with an independent $N_{\text{state}}$-by-$M$ Q-table, where $N_{\text{state}}$ is the total number of possible states and $M$ is the number of action options. Each mentor is assigned one of seven predefined artificial strategies. In each iteration, we randomly select an agent $p_1$ and player $p_2$ from the pool of agents and mentors to participate in an $L$-round repeated Prisoner's Dilemma game. In the $t$-th round, each agent assesses the current state $s_{p_1,t}$ (i.e., the action pairs from the last $\ell$ interactions) and consults its Q-table to choose an action. Agents employ an $\epsilon$-greedy strategy for action selection, effectively balancing exploitation and exploration. Mentors follow their predefined strategies. After $L$ rounds of interactions, the rewards $W_{p_1}(s_{p_1,t}, a_{p_1,t})$ are calculated for each action decision in the $L$-round game, based on the agent's payoffs and the interaction outcomes. Subsequently, each agent's Q-table is updated using the Bellman equation (see Eq. (4)).

$W_{p_X}(s_{p_X,t}, a_{p_X,t})$ to player $p_X$'s decision $a_{p_X,t}$ in state $s_{p_X,t}$, given by

$$W_{p_X}(s_{p_X,t}, a_{p_X,t}) = \begin{cases} \theta U_{p_X,t} + (1-\theta)\overline{U}_{p_X} & if\ \overline{U}_{p_X} \geq \overline{U}_{p_X}, \\ \theta U_{p_X,t} & otherwise. \end{cases} \quad (3)$$

Above $p_{-X}$ denotes the opponent of player $p_X$, such as $p_2$ in relation to $p_1$. The Q-table for $p_X$ (denoted by $\mathbf{Q}_{p_X}$) is then updated using the Bellman equation:

$$NewQ_{p_X}(s_{p_X,t}, a_{p_X,t}) = Q_{p_X}(s_{p_X,t}, a_{p_X,t}) + \alpha \Big[ W_{p_X}(s_{p_X,t}, a_{p_X,t}) + \gamma \max_{a'} Q_{p_X}(s', a') - Q_{p_X}(s_{p_X,t}, a_{p_X,t}) \Big], \quad (4)$$

where $\alpha$ represents the learning rate, and $0 \leq \gamma < 1$ is the discount factor. The term $\max_{a'} Q_{p_X}(s', a')$ reflects the estimated maximum future reward from the next state $s'$, highlighting the importance of future rewards in decision-making. For a detailed description of our method, refer to the "pseudocode" provided in Methods. Multi-agent Q-learning has been proven effective in complex strategic interactions. Our approach extends this framework by introducing a reward mechanism that balances defeating the opponent and maximizing individual payoffs. This extension, parameterized by $\theta$ as described in Eq. (3), allows for adaptive strategies in games such as the Prisoner's Dilemma.

The training environment comprises $N_a = 49$ agents with blank strategies and $N_m = 49$ mentors implementing predefined strategies, including TFT, GTFT, WSLS, Hold-a-Grudge, Fool-Me-Once, OmegaTFT, and GradualTFT. Each category of predefined artificial strategies is used by seven mentors, creating a diverse and comprehensive training environment. Notably, both OmegaTFT and GradualTFT extend beyond simple one-step memory strategies, incorporating more complex, manually designed strategies. During training, we use a special Prisoner's Dilemma parameter set: $R = 2$, $S = 0$, $T = 3$, and $P = 0.1$. This choice is deliberate, as the lower payoff for mutual defection ($P = 0.1$) reduces the appeal of defection, fostering an environment more conducive to the emergence of cooperative behavior. The goal is to encourage agents to learn and adopt cooperative strategies. The learning rate $\alpha$ is set to 0.2, the discount factor $\gamma$ for future considerations is set to 0.5, and the preference for scores $\theta$ is set to 0.8. These parameters were selected to balance the speed of learning,

future reward valuation, and preference for achieving high payoffs during training.

## Pseudocode

This section provides the detailed pseudocode for the multi-agent Q-learning algorithm discussed in the main text. Algorithm 1 is central to our study, which aims to explore strategic dynamics in Markov games under general-sum conditions. The pseudocode elaborates on the iterative process used by each agent to update their respective Q-values based on their experiences and interactions with other agents and mentors in a simulated environment. It outlines the steps each agent undergoes during the training phase of the evolutionary repeated Prisoner's Dilemma game, including decision-making processes, strategy updates, and reward calculations. These components collectively drive the learning and adaptation of strategies over time. The pseudocode is designed to provide clarity on the computational mechanics underlying our approach, offering a comprehensive view of the procedural steps involved.

ensures a comprehensive training environment for our learning agents. The details of each strategy are as follows:

1. Tit-for-Tat (TFT): This strategy starts with cooperation and then replicates the opponent's previous move. Known for its simplicity and effectiveness, TFT encourages mutual cooperation.
2. Generous-Tit-for-Tat (GTFT0.3): A variant of TFT, GTFT0.3 forgives defections 30% of the time. When the opponent defects, GTFT0.3 will cooperate with a 30% probability in the next move, unlike TFT, which never cooperates after a defection.
3. Win-Stay-Lose-Shift (WSLS): This strategy repeats the previous action if it was "successful" (yielding $R$ or $T$ payoffs) and switches actions if it was "unsuccessful" (yielding $S$ or $P$ payoffs).
4. Hold-a-Grudge: Once the opponent defects, this strategy perpetually defects in all subsequent rounds, never forgiving the initial defection[28].
5. Fool-Me-Once: This strategy issues a warning by defecting once if the opponent defects but forgives and cooperates again if the opponent cooperates. However, upon a second defection by the opponent, it defects indefinitely[28].

---

**Algorithm 1**. **Multi-Agent Q-Learning for Exploring Dominant Strategies in Iterated and Evolutionary Games**

 **Input:** Number of agents $N_a$, number of mentors $N_m$, number of training iterations $I_{total}$, memory length $\ell$, number of actions $M$, number of rounds per game $L$, learning rate $\alpha$, discount factor $\gamma$, preference for scores $\theta$, exploration rate $\epsilon$

 **Output:** Optimized Q-tables for each agent

1: Calculate $N_{state} = \left(\frac{M^{2\ell+2}-1}{M^2-1}-1\right)$ ▷ Compute the total number of possible states

2: Initialize $N_a$ independent $N_{state}$-by-$M$ Q-tables for all agents with arbitrary values

3: **for** $i = 1$ to $I_{total}$ **do** ▷ Each iteration is one training episode

4: Randomly select an individual $p_1$ from $N_a$ agents

5: Randomly select an individual $p_2$ from the pool of $N_a$ agents and $N_m$ mentors

6: **for** $t = 1$ to $L$ **do** ▷ Run each repeated game for $L$ rounds

7: $p_1$ determines its current state $s_{p_1,t}$ based on the last $\ell$ actions of itself and $p_2$

8: $p_1$ chooses action $a_{p_1,t}$ using the $\epsilon$-greedy policy from its Q-table $Q_{p_1}(s,a)$

9: **if** $p_2$ is an agent **then**

10: $p_2$ determines its current state $s_{p_2,t}$ and chooses action $a_{p_2,t}$ similarly

11: **else**

12: $p_2$ chooses action $a_{p_2,t}$ based on its predefined mentor strategy

13: **end if**

14: Both individuals execute their actions $a_{p_1,t}$ and $a_{p_2,t}$

15: Record payoffs $U_{p_1,t}$ and $U_{p_2,t}$ for $p_1$ and $p_2$ respectively

16: **end for**

17: Calculate the average payoff $\overline{U}_{p_1} = (\sum_{t=1}^{L} U_{p_1,t})/L$ and $\overline{U}_{p_2} = (\sum_{t=1}^{L} U_{p_2,t})/L$

18: **for** $t = 1$ to $L$ **do** ▷ Compute reward for each round (state)

19: Compute reward $W_{p_1}(s_{p_1,t}, a_{p_1,t})$ for $p_1$ using Eq. (3) from the model description

20: **if** $p_2$ is an agent **then**

21: Compute reward $W_{p_2}(s_{p_2,t}, a_{p_2,t})$ for $p_2$ using Eq. (3)

22: **end if**

23: **end for**

24: **for** $t = 1$ to $L$ **do** ▷ Update Q-values using the Bellman equation

25: Update $p_1$'s Q-table using the Bellman equation (Eq. (4) from the model description)

26: **if** $p_2$ is an agent **then**

27: Update $p_2$'s Q-table using the Bellman equation

28: **end if**

29: **end for**

30: **end for**

---

## Mentor strategies

In our study, we use seven distinct strategies as mentors to guide and evaluate the learning agents in the evolutionary repeated Prisoner's Dilemma. Each mentor strategy begins with a random choice between cooperation and defection on the first move. These mentor strategies provide a diverse set of behavioral models, ranging from simple reciprocal strategies to more complex, adaptive approaches. This variety

6. Omega-Tit-for-Tat (OmegaTFT): We use the common configuration of OmegaTFT with parameters "OmegaTFT: 3,8". OmegaTFT behaves like TFT but includes deadlock and randomness detection:
   - Deadlock Detection: If the pattern of alternating between cooperation and defection occurs more than three times consecutively (i.e., the sequence "cooperate-defect-cooperate-

defect" is observed), OmegaTFT will cooperate to break the deadlock and reset the count.

- Randomness Detection: Randomness increases by 1 if the opponent's current move differs from its previous move or if the opponent's move differs from OmegaTFT's move. Randomness decreases by 1 if the opponent's current move matches its previous move. If randomness reaches 8, OmegaTFT will defect indefinitely to counter perceived randomness[29].

- OmegaTFT combines adaptability and deadlock detection, allowing it to break out of mutual defection cycles and seek cooperative opportunities.

7. Gradual-Tit-for-Tat (GradualTFT): This strategy cooperates if the opponent cooperates. If the opponent defects, GradualTFT defects for $N_D$ rounds, where $N_D$ is the total number of defections by the opponent. After retaliating, GradualTFT enters a two-round "cooling-off" period, cooperating twice regardless of the opponent's actions. GradualTFT emphasizes gradual retaliation to deter further defections while maintaining the potential for future cooperation[30].

These mentor strategies provide a diverse set of behavioral models for the agents to learn from and interact with. TFT, GTFT0.3, and WSLS represent foundational strategies in the study of the repeated Prisoner's Dilemma, while Hold-a-Grudge and Fool-Me-Once introduce more punitive approaches. OmegaTFT and GradualTFT offer sophisticated mechanisms for enhancing cooperation and navigating complex strategic interactions. This variety ensures a comprehensive training environment, promoting the development of robust and adaptive agent strategies.

## Zero-determinant strategies

To evaluate the learning agents under more controlled payoff relationships, we introduce a set of eight Zero-Determinant (ZD) strategies based on the foundational work of Press and Dyson[21]. ZD strategies enforce a linear relation between the player's own payoff and that of the opponent, of the form $\pi_{\text{opponent}} = l + s(\pi_{\text{self}} - l)$, where $l$ is the baseline payoff, and $s$ is the slope of the enforced linear relationship. By carefully choosing the parameters $\phi$, $s$, and $l$, ZD strategies can take on different behavioral roles, including extortionate, generous, and mischief-like behavior. The details of each ZD strategy are as follows:

1. **ZDExtort2**: An extortionate strategy with $l = P$, $s = 0.5$, and $\phi = \frac{1}{9}$. It enforces a payoff relation in which the opponent gains only a fraction of any surplus beyond $P$ achieved by the ZD player[22].

2. **ZDExtort2v2**: A variant of ZDExtort2 with a slightly larger $\phi = \frac{1}{8}$, using the same slope $s = 0.5$ and baseline payoff $l = P$. The increase in $\phi$ alters the cooperation probabilities while preserving the extortionate nature of the strategy.

3. **ZDExtort3**: An extortionate strategy with a steeper extortion factor ($s = \frac{1}{3}$, $\phi = \frac{3}{26}$, and $l = P$), making it more demanding and aggressive in enforcing its advantage over the opponent[21].

4. **ZDExtort4**: A more extreme extortion strategy with $s = 0.25$, $\phi = \frac{4}{17}$, and $l = P$. This configuration strongly favors the ZD player and leaves minimal room for opponent benefit beyond the baseline payoff[69].

5. **ZDGen2**: A generous ZD strategy with $l = R$, $s = 0.5$, and $\phi = \frac{1}{8}$, promoting mutual cooperation by allowing the opponent to benefit more than the ZD player when cooperation is sustained.

6. **ZDGTFT2**: Another generous variant with $l = R$, $s = 0.5$, and a larger $\phi = 0.25$, behaving similarly to GTFT but derived analytically from the ZD framework[22].

7. **ZDMischief**: A mischief-type strategy with $l = P$, $s = 0$, and $\phi = 0.1$, aiming to keep the opponent's payoff fixed regardless of their actions. While mathematically extreme, it highlights the versatility of ZD configurations.

8. **ZDSet2**: A ZD strategy with intermediate generosity, using $l = \frac{R+P}{2}$, $s = 0$, and $\phi = \frac{1}{4}$. This player fixes the opponent's payoff around the midpoint between cooperation and punishment outcomes.

These ZD strategies span a wide range of behavioral archetypes-from exploitative extortion to cooperative generosity to payoff fixation.

## Software environment

All simulations and analyses were performed using Python (version 3.8.5) and Julia (version 1.9.4). Python computations relied on NumPy (version 1.24.4), Matplotlib (version 3.7.5), and the Python standard library modules sys, random, time, json, collections, pickle, and datetime. Julia computations used the packages ArgParse (version 1.2.0), DataFrames (version 1.6.1), Distributions (version 0.25.108), IterativeSolvers (version 0.9.4), and StatsBase, together with the standard libraries LinearAlgebra, Random, SparseArrays, and Printf.

## Replicator dynamics

We consider the evolutionary dynamics of $n$ strategies (labeled in $1, \cdots, n$) in a well-mixed population of $N$ players, with payoff matrix

$$
\begin{pmatrix}
a_{11} & a_{12} & \cdots & a_{1n} \\
a_{21} & a_{22} & \cdots & a_{2n} \\
\vdots & \vdots & \ddots & \vdots \\
a_{n1} & a_{n2} & \cdots & a_{nn}
\end{pmatrix}, \tag{5}
$$

where $a_{ij}$ denotes the payoff to an individual when adopting strategy $i$ against the opponent using strategy $j$. Define the number of $i$-individuals as $X_i$ and the state vector $\mathbf{X} = (X_1, X_2, \cdots, X_n)$. Consequently, the average payoff for individuals using strategy $i$ is obtained

$$
\overline{U}_i(\mathbf{X}) = \frac{1}{N-1}(a_{i1}X_1 + \cdots + a_{ii}(X_i - 1) + \cdots + a_{in}X_n). \tag{6}
$$

Player $i$ imitates player $j$'s strategy with probability

$$
p_{i \to j} = \frac{1}{1 + \exp(\delta(\overline{U}_i - \overline{U}_j))}. \tag{7}
$$

The stochastic evolution process can be formulated in terms of the master equation

$$
P^{\tau+1}(\mathbf{X}) - P^\tau(\mathbf{X}) = \sum_{i=1}^{n} P^\tau(\mathbf{X} - \mathbf{e}_i)T^+(\mathbf{X} - \mathbf{e}_i) + \sum_{i=1}^{n} P^\tau(\mathbf{X} + \mathbf{e}_i)T^-(\mathbf{X} + \mathbf{e}_i)
$$
$$
- \sum_{i=1}^{n} P^\tau(\mathbf{X})T^+(\mathbf{X}) - \sum_{i=1}^{n} P^\tau(\mathbf{X})T^-(\mathbf{X}), \tag{8}
$$

where $P^\tau(\mathbf{X})$ is the probability that the population is in state $\mathbf{X}$ at time $\tau$ and $\mathbf{e}_i$ is the vector with 1 in the $i_{\text{th}}$ position and 0 elsewhere. Introducing the notation $x_i = X_i/N$, $\mathbf{x} = \mathbf{X}/N$, $t = \tau/N$, and the probability density function $\rho(\mathbf{x}, t) = NP^\tau(\mathbf{X})$ yields

$$
\rho(\mathbf{x}, t + N^{-1}) - \rho(\mathbf{x}, t) = \sum_{i=1}^{n} \rho(\mathbf{x} - \mathbf{e}_i/N, t)T_i^+(\mathbf{x} - \mathbf{e}_i/N, t)
$$
$$
+ \sum_{i=1}^{n} \rho(\mathbf{x} + \mathbf{e}_i/N, t)T_i^-(\mathbf{x} + \mathbf{e}_i/N, t) \tag{9}
$$
$$
- \sum_{i=1}^{n} \rho(\mathbf{x}, t)T_i^-(\mathbf{x}) - \sum_{i=1}^{n} \rho(\mathbf{x}, t)T_i^+(\mathbf{x}).
$$

The probability density function and the transition probability can be expanded in a Taylor series at $(\mathbf{x}, t)$ for large $N$. Neglecting high order terms in $N^{-1}$, we get

$$\frac{\partial}{\partial t}\rho(\mathbf{x}, t) = -\sum_{i=1}^{n}\frac{\partial}{\partial x_i}(\phi_i(\mathbf{x})\rho(\mathbf{x}, t)) + \sum_{i=1}^{n}\frac{\partial^2}{\partial x_i^2}(\psi_i^2(\mathbf{x})\rho(\mathbf{x}, t)), \quad (10)$$

under the condition of weak selection, where

$$\phi_i(\mathbf{x}) = T_i^+(\mathbf{x}) - T_i^-(\mathbf{x}) = \frac{\delta N}{2(N-1)}x_i(\overline{U}_i - \overline{U}), \quad (11a)$$

$$\psi_i(\mathbf{x}) = \sqrt{(T_i^+(\mathbf{x}) + T_i^-(\mathbf{x}))/N} = \sqrt{x_i(1-x_i)/(N-1)}. \quad (11b)$$

The partial differential equation has the form of a Fokker-Planck equation. Meanwhile, we can derive the corresponding Langevin equation

$$\dot{x}_i = \phi_i(\mathbf{x}) + \psi_i(\mathbf{x})\xi, \quad (12)$$

where $\xi$ is the Gaussian noise. For a sufficiently large population size, the influence of noise becomes negligible, leading to the replicator equations:

$$\dot{x}_i = \frac{\delta N}{2(N-1)}x_i(\overline{U}_i - \overline{U}), \quad (13)$$

for all $i \in \{1, 2, \cdots, n\}$.

### Reporting summary
Further information on research design is available in the Nature Portfolio Reporting Summary linked to this article.

## Data availability
All results are based on numerical simulations. Data underlying all main-text and Supplementary Figs. are available in a public repository and its archived snapshot (GitHub: https://github.com/YuzukiWang/multiagent_q_learning_for_evolutionary_games; Zenodo: https://doi.org/10.5281/zenodo.17562512).

## Code availability
All code used in this study, implemented in Python and Julia, including the multi-agent reinforcement-learning framework used to discover the memory-two bilateral reciprocity (MTBR) strategy, the simulation code used to evaluate strategies and generate the numerical simulation outputs reported in the main text and Supplementary Information, and all analysis and plotting scripts and configuration files required to reproduce the reported results from the provided data, is openly available on GitHub: https://github.com/YuzukiWang/multiagent_q_learning_for_evolutionary_games. For citability and reproducibility, the version of the repository associated with this study has been archived on Zenodo[70] (https://doi.org/10.5281/zenodo.17562512).

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

## Acknowledgements

Q.S. acknowledges support from the National Natural Science Foundation of China (No. 62473252) and the State Key Laboratory of Autonomous Intelligent Unmanned Systems (No. ZZKF2025-1-4). L.W. acknowledges support from the National Natural Science Foundation of China (No.62533002, No. 62036002).

## Author contributions

Q.S., H.W. and L.W. conceived and designed the research. Q.S., H.W. and L.W. contributed to the methodology development. H.W. conducted the data analysis. H.W. and Y.X. performed the theoretical analysis and data visualization. Q.S. and H.W. drafted the first version of the manuscript. Q.S., H.W., L.W., and Y.X. contributed to the interpretation of the results, critical revision of the manuscript, and approval of the final version of the manuscript. L.W. supervised the project. Q.S. and H.W. contributed equally to the work.

## Competing interests

The authors declare no competing interests.

## Additional information

Qi Su or Long Wang.

**Peer review information** *Nature Communications* thanks Zhixue He and
the other, anonymous, reviewer(s) for their contribution to the peer
review of this work. A peer review file is available.

