## [Transparent Peer Review file · Nature Communications]

A multi-agent reinforcement learning framework for exploring dominant strategies in iterated and evolutionary games

Corresponding Author: Professor Long Wang

Version 0:

Reviewer comments:

Reviewer #1

(Remarks to the Author)

Exploring dominant strategies in iterated games is of great theoretical and practical importance across various fields. Previous research has uncovered classic strategies like tit-for-tat and generous-tit-for-tat through mathematical analysis of limited cases. These strategies provide valuable insights into human decision-making but represent only a fraction of potential strategies due to the constraints imposed by the mathematical and computational tools previously available to explore extensive strategy spaces.

To address this limitation, this study introduces a novel approach using multi-agent reinforcement learning to investigate complex decision-making processes that extend beyond human intuition. This method has led to the discovery of a new strategy, termed memory-two bilateral reciprocity. This strategy consistently outperforms a wide range of existing strategies in pairwise interactions and achieves high payoffs, suggesting a significant advancement in understanding strategic interactions.

When this new strategy is introduced into evolving populations featuring a diversity of strategies, it exhibits dominance and significantly enhances levels of cooperation and social welfare. This is observed in both homogeneous and heterogeneous population structures and across different types of games. The effectiveness of memory-two bilateral reciprocity is confirmed through both simulations and mathematical analysis. This research underscores the capability of multi-agent reinforcement learning to unveil dominant strategies in complex settings, providing a fresh perspective on strategy exploration in iterated games and expanding our understanding of strategic dynamics.

I have very much enjoyed reading this manuscript. I find it comprehensive and reporting important new and relevant results that will surely also inspire further research along similar lines. My recommendation is therefore only minor revision. The following comments should please be considered.

1. Background and Literature Review: The introduction and background sections could benefit from a more detailed discussion of the connections between classic studies in game theory and reinforcement learning, particularly their relevance to the proposed approach. Expanding the discussion to include more recent work in the fields of iterated games and strategy evolution would enrich the context. Reinforcement learning, specifically, has been considered in *Knowl.-Based Syst.* 301, 112326 (2024) and *New J. Phys.* 23, 083020 (2021), for example.

2. Pseudocode Improvement: The pseudocode section could be revised to make it more accessible to a broader audience. Adding more comments and simplifying certain steps would help readers from less technical backgrounds better understand the algorithm. This would also improve reproducibility and comprehension for researchers seeking to build upon your work.

3. Discussion on Noise: In the section "The dominance of MTBR in evolving populations," the results are presented under noise-free conditions. Extending this discussion to include the effects of noise on evolutionary trajectories and the dominance of MTBR would make the analysis more comprehensive. The importance of noise for evolutionary outcomes has been studied in *New J. Phys.* 8, 22 (2006), and in many other publications in the past decades.

4. Terminology Consistency: The term "dominant strategies" is used frequently throughout the manuscript but is not consistently defined. It is recommended to clarify whether it refers to payoff superiority, evolutionary prevalence, or strategic

stability. Providing a clear and consistent definition would avoid ambiguity and enhance the clarity of your results and conclusions.

If a revision is granted, and if needed, I will be happy to review the manuscript again.

(Remarks on code availability)

Reviewer #2

(Remarks to the Author)

This study explores a classical topic: What constitutes a "strong" strategy in the Iterated Prisoner's Dilemma (IPD)? Using multi-agent reinforcement learning, the authors identified a strategy called memory-two bilateral reciprocity (MTBR). This strategy behaves like tit-for-tat (TFT) in most cases but cooperates when both players defect twice in a row as shown in the Supplementary Table. Using MTBR, the authors analyzed its performance in tournaments and evolutionary dynamics.

The authors found that MTBR performs well in these scenarios and claim that introducing MTBR into a population increases the overall level of cooperation.

However, after a careful review of the manuscript, I cannot recommend it for publication. My decision is based on several critical concerns, the most significant of which is the authors' arbitrary selection of strategies (based on their subjective choice of "well-recognized strategies") as potential opponents for MTBR. This selection applies not only to the training process but also to the tournament and evolutionary dynamics. Such an approach introduces significant bias and limits the generality of the findings. For instance, MTBR would likely be exploited by the ALLD strategy, which is not adequately addressed.

Moreover, the authors should analyze MTBR's properties using more rigorous and objective methods. For example, is MTBR a partner strategy for certain benefit-to-cost (b/c) ratios? [1] Is it an evolutionarily robust strategy? [2]? These questions are fundamental for establishing the robustness of a strategy in the IPD. Established theoretical frameworks exist to address such questions, and incorporating these would greatly strengthen the research. Examples include:

> [1] Hilbe, Christian, Krishnendu Chatterjee, and Martin A. Nowak. "Partners and rivals in direct reciprocity." *Nature human behaviour* 2.7 (2018): 469-477.

> [2] Stewart, Alexander J., and Joshua B. Plotkin. "From extortion to generosity, evolution in the iterated prisoner's dilemma." *Proceedings of the National Academy of Sciences* 110.38 (2013): 15348-15353.

Additionally, the manuscript overlooks a substantial body of prior research on longer-memory strategies. While many studies focus on memory-one strategies, there is also extensive work on longer-memory strategies, which the authors fail to acknowledge. A meaningful comparison with this literature is essential to contextualize the findings. For instance:

> All or None strategies: Hilbe, Christian, et al. "Memory-n strategies of direct reciprocity." *Proceedings of the National Academy of Sciences* 114.18 (2017): 4715-4720.

> TFT-ATFT : Do Yi, Su, Seung Ki Baek, and Jung-Kyoo Choi. "Combination with anti-tit-for-tat remedies problems of tit-for-tat." *Journal of Theoretical Biology* 412 (2017): 1-7.

> CAPRI : Murase, Yohsuke, and Seung Ki Baek. "Five rules for friendly rivalry in direct reciprocity." *Scientific reports* 10.1 (2020): 16904.

> cumulative reciprocity : Li, Juan, et al. "Evolution of cooperation through cumulative reciprocity." *Nature Computational Science* 2.10 (2022): 677-686.

> evolution in longer-memory strategy space: Murase, Yohsuke, and Seung Ki Baek. "Grouping promotes both partnership and rivalry with long memory in direct reciprocity." *PLOS Computational Biology* 19.6 (2023): e1011228.

> longer-memory partner strategies : Glynatsi, Nikoleta E., et al. "Conditional cooperation with longer memory." *Proceedings of the National Academy of Sciences* 121.50 (2024): e2420125121.

Furthermore, the advantage of MTBR over other strategies appears marginal. As shown in Figure 3, its performance is comparable to GTFT. Similarly, the claim in Figure 4c that introducing MTBR improves the overall cooperation level is not compelling, as the improvement is only 0.03. The visual presentation exaggerates the significance of this result.

While these are not exhaustive concerns, the critical issues raised led me to conclude that the manuscript is not suitable for publication.

(Remarks on code availability)

Reviewer #3

(Remarks to the Author)

In this study, the authors investigate dominant strategies in repeated games using a reinforcement learning framework. Their work provides a novel approach to analyzing dominant strategies and reports an intriguing new strategy—Memory-Two Bilateral Reciprocity (MTBR). The authors demonstrate that MTBR stands out among various strategies, facilitating mutual

cooperation and enhancing overall payoffs. Moreover, they employ an evolutionary model to study the evolutionary stability of this strategy. I appreciate the introduction of this innovative framework and the discovery of a new, special strategy, which contribute to advancing research in evolutionary game theory and cooperation dynamics. This is very interesting and meaningful work; However, before I agree to accept it, the authors could benefit from clarifying certain aspects. Below are my comments:

The authors could benefit from providing a more detailed explanation of the design of agents' action objectives. Within the reinforcement learning framework, an agent's actions are fundamentally guided by the goal of maximizing an objective or utility function (i.e., here W_{PX}), which the authors consider as the relative advantage over opponents and their own payoffs. I would appreciate further clarification on the motivation and rationale behind this design of W_{PX} .

How are the initial actions of the RL-based agents determined? Do they randomly choose between cooperation and defection? Additionally, does the choice of initial action affect the results? For example, certain strategies, such as the Grim strategy, are particularly sensitive to initial actions in interactions. Specifically, after each training step (L steps), when individuals are rematched to play the game, how is the initial strategy defined in this context?

During the training process, does the probability of an agent encountering a mentor (which is equivalent to the number of mentors) lead to different training outcomes? Additionally, does it affect the convergence of the RL training?

I noticed that the authors mention, "Our goal is to explore strategies that not only dominate the opponent but also coordinate for a high collective payoff." Does this imply an exploration of prosocial or other-regarding strategies? Given that recent studies have discussed and emphasized human prosocial preferences [e.g., PNAS 2024, 121(49): e2412195121], could the authors provide some discussion on this aspect?

The specific definition of the state in the model is unclear. Could the authors further clarify what the elements in the state set represent? Stating this in the main text could help readers intuitively understand the concept without relying on supplementary materials. An example might be useful to illustrate this point.

I noticed that the authors use a specific payoff matrix. I would like to understand whether the results are dependent on a specific particular matrix. Recent studies have introduced the concept of dilemma strength in the Prisoner's Dilemma [e.g., Physics of life reviews, 2015, 14: 1-30 (2015)], providing a quantifiable standard across the payoff matrix. This allows research on the Prisoner's Dilemma to be independent of specific payoff matrices. Could the authors include related results or discussions on this? This might enhance the generalizability of the findings.

In the results section, the authors mention the "prosperous state." Could the authors clarify what this refers to and explain the criteria used to determine it?

In "The dominance of MTBR in evolving populations", the authors combine the MTBR strategy they discovered with an evolutionary model to derive results. I would like to understand how the MTBR strategy is defined within this model. Does it refer to individuals using a pre-trained q -table for decision-making interactions? Additionally, does the model account for mutations? Some literature has also emphasized the importance of mutations in evolutionary models [e.g., PNAS, 2023, 120(20): e2221080120].

(Remarks on code availability)

The provided code is thorough and includes clear instructions for running the application.

Reviewer #4

(Remarks to the Author)

In premising repeated game situations, the authors found a new strategy, named by memory-two bilateral reciprocity (MTBR) strategy by means of Q-learning framework, one of the most commonly accepted schemes of reinforcement learning, which wouldn't be detected by human intuition. This approach seems much smarter than usual simple applications of the reinforcement learning that are rather boring.

The authors carefully explored a series of simulations to prove that the existence of their new strategy; MTBR, drives up the payoffs of other entire strategies that are commonly applied such as TFT, PAVLOV (Win-stay & Lose-shift), ZD strategies; Fig. 3. This is very persuasive.

In addition, the authors explored what MTBR brings in a spatial game setting over entire symmetric 2 by 2 game classes; not only PD, but also, Chicken (Snowdrift), Stag Hunt, and Trivial games. Quite interestingly, MTBR loses dominant power over other conventional strategies as above in the region of $S+T>2R$.

I think this is a very important finding amid the authors' reports.

As the authors partially discussed, this is because, in the region, just simply mutual cooperating does not obtain a socially optimal (SO) situation due to the so-called ST-reciprocity effectively working beyond R-reciprocity. To emerge such strategies delivering ST-reciprocity, a bit longer memory help might be requisite like a sort of correlation strategies can work. Needless to say, R-reciprocity means that taking mutual cooperation ensures a SO avoiding mutual defection. While, ST-reciprocity indicates altering T (D-C) and S (C-D) in the time direction can deliver a SO.

The authors should supplement their discussion by referring what I abovementioned and cite relevant literatures about R- and ST-reciprocity; (i) Direct reciprocity in spatial populations enhances R-reciprocity as well as ST-Reciprocity, PLOS One

8 (8), e71961, 2013, (ii) A study on emergence of alternating reciprocity in a 2x2 game with 2-length memory strategy. BioSystems 90: 728–737.

As a whole, I have a positive feeling on the MS...

(Remarks on code availability)

Version 1:

Reviewer comments:

Reviewer #1

(Remarks to the Author)

The authors have revised their manuscript comprehensively and with love to detail. I warmly recommend publication in present form.

(Remarks on code availability)

Reviewer #2

(Remarks to the Author)

While I appreciate the substantial effort that the authors have made to improve the manuscript, I don't recommend publication of this manuscript since the critical issues remain.

First and foremost, as I mentioned in my previous comment, the authors evaluated the strategy against a specific choice of the strategy set. While they introduced another set of strategies that indeed partially addresses the issue, the concern I raised remains unresolved.

I'm not criticizing the issue in MTBR itself, but rather the methodology employed by the authors in assessing the strategy. Every strategy has its own strengths and weaknesses, and the authors failed to present these objectively. For instance, they should have fairly compared the strategy with strategies in a given complexity class, such as memory-1 or memory-2 mixed strategies.

For example, if the population predominantly consists of ALLD, MTBR would likely fail miserably. Again, I'm not suggesting that the strategy needs to be robust against ALLD (which is obviously impossible). Instead, I'm emphasizing that the success of a strategy depends on the specific environmental setting we assumed. We need to demonstrate when the strategy performs well and when it fails.

In the revised manuscript, the authors presented a more robust analysis to determine whether the strategy is a partner strategy or not and to assess its evolutionary robustness. I commend this direction. However, despite their claim, MTBR is not a partner strategy. I calculated the stationary distribution using Eq.(8) in the rebuttal letter and found that the strategy fails to meet the first condition of being a partner: it is not fully self-cooperative under infinitesimal error.

To my understanding, the reason why MTBR appears successful in their experiments is due to the following factors. Firstly, they eliminated execution error entirely while assuming that the initial move is randomly chosen, which is a rather specific assumption. (While Fig.R1 demonstrates their study of the case with noise, those instances are relatively rare since they only continue the game for 20 rounds.) Secondly, most of the strategies they considered maintain cooperation if the initial moves in the first few rounds were successful. MTBR is designed to excel in the initial rounds with those specific opponent strategies. Additionally, they primarily focus on repeated games with a length of 20 rounds, which strikes a balance between being neither too short nor too long. If the length is shorter, defecting strategies are more likely to be chosen. Conversely, if the length is longer (such as 10^3 or 10^4), the advantage of the initial moves diminishes, and they would exhibit similar performance to other strategies.

Considering those, the scientific significance of the work is limited. It does not appear to offer substantial new insights beyond existing literature.

(Remarks on code availability)

Reviewer #3

(Remarks to the Author)

I appreciate the authors' revisions and efforts. All my concerns have been addressed, I have no further comments and recommend acceptance.

(Remarks on code availability)

Reviewer #4

(Remarks to the Author)

The revised MS is good enough to persuade me evaluating that this version can be acceptable to the journal. It sincerely responds all the points and suggestions the reviewers gave. I'm glad to draw a positive evaluation on the MS.

(Remarks on code availability)

Version 2:

Reviewer comments:

Reviewer #2

(Remarks to the Author)

The authors have addressed many of the concerns raised in my previous review and have improved the manuscript by reporting the limitations of the proposed strategy in more objective ways and providing additional analysis. I appreciate the clarification regarding my misunderstanding of the definition of partner strategies, and I understand that the paper only considers the error-free case.

While I acknowledge the authors' effort, several critical concerns remain unsolved.

- As the authors mentioned, they consider only the error-free case. (Although they present some results with noise, the main focus is on the error-free case.) Why then does canonical TFT or GRIM not perform well in this setting? As I pointed out in my previous review, the success of MTBR (unsuccessfulness of TFT and GRIM) is likely due to the specific choice of the initial condition. While the authors are focusing on the error-free case, they assume random initial moves, which seems quite inconsistent. If we consider the error-free and finite-rounds games, it is natural to assume that both players start with action prescribed by the strategy. In that case, I suspect that even canonical TFT or GRIM would perform as well as MTBR. As shown in Table 1, the prescription of MTBR is almost identical to that of TFT, with the only difference being in the actions for the mutual defections in the last two rounds. MTBR's apparent success appears to hinge on the assumption that the initial moves are random. I don't think MTBR is particularly superior to TFT as their moves are almost identical.

- The amount of improvement that MTBR causes (Fig. 4c, 4f, and also seen in Fig. 3) is not significant. As I pointed out in my first comment and they replied and revised in their second submission, the amount of the improvement is very small (2.9 -> 2.938) although it visually looks significant in their plots, (which I find rather misleading.) They defended their argument claiming that the theoretical maximum is 3.0 and the improvement is 38% of the theoretical limit. However, this is a nonsense argument. If the improvement were 2.98 -> 2.99, the improvement would be 50% of the theoretical limit and even more significant according to their argument. Of course, such an improvement would be insignificant.

- The dominance of the MTBR in evolutionary settings may be significantly enhanced in their evolutionary simulations (Fig. 4b). As we see in Fig.3, the payoffs of MTBR and some of the other strategies (such as GTFT) are similar. Nevertheless, in Fig.4b, MTBR looks outstanding. Once the other aggressive strategies are wiped out and a cooperative environment is established, what matters is the self-cooperation levels. Even if the advantage of MTBR's self-cooperation level is minor (according to Supplementary Figure 2, MTBR's self-payoff is 2.94 while that for Gradual TFT is 2.91), such an advantage may look significantly enhanced in their settings, possibly due to large N.

- Actually, the above two also hinge on their settings that "the initial round is random while the other rounds are completely error-free", which I pointed out as my main concern. As far as I understand, MTBR is a kind of hand-shaking strategy. It behaves like TFT but does well in their random initial configuration. If the action in the initial round were fixed (which is a standard setting in finite-rounds games), other strategies would perform equally well and the apparent advantage of MTBR may disappear.

In my opinion, those points are enough to draw a conclusion. I wouldn't be able to recommend publication even if it were submitted to other journals than Nature Comm.

(Remarks on code availability)

Replies to comments and suggestions from referee 1

Comments:

Exploring dominant strategies in iterated games is of great theoretical and practical importance across various fields. Previous research has uncovered classic strategies like tit-for-tat and generous-tit-for-tat through mathematical analysis of limited cases. These strategies provide valuable insights into human decision-making but represent only a fraction of potential strategies due to the constraints imposed by the mathematical and computational tools previously available to explore extensive strategy spaces.

To address this limitation, this study introduces a novel approach using multi-agent reinforcement learning to investigate complex decision-making processes that extend beyond human intuition. This method has led to the discovery of a new strategy, termed memory-two bilateral reciprocity. This strategy consistently outperforms a wide range of existing strategies in pairwise interactions and achieves high payoffs, suggesting a significant advancement in understanding strategic interactions.

When this new strategy is introduced into evolving populations featuring a diversity of strategies, it exhibits dominance and significantly enhances levels of cooperation and social welfare. This is observed in both homogeneous and heterogeneous population structures and across different types of games. The effectiveness of memory-two bilateral reciprocity is confirmed through both simulations and mathematical analysis. This research underscores the capability of multi-agent reinforcement learning to unveil dominant strategies in complex settings, providing a fresh perspective on strategy exploration in iterated games and expanding our understanding of strategic dynamics.

I have very much enjoyed reading this manuscript. I find it comprehensive and reporting important new and relevant results that will surely also inspire further research along similar lines. My recommendation is therefore only minor revision. The following comments should please be considered.

We thank the reviewer for the thorough review and thoughtful suggestions. We truly appreciate the insights and all the constructive suggestions below.

Background and Literature Review: The introduction and background sections could benefit from a more detailed discussion of the connections between classic studies in game theory and reinforcement learning, particularly their relevance to the proposed approach. Expanding the discussion to include more recent work in the fields of iterated games and strategy evolution would enrich the context. Reinforcement learning, specifically, has been considered in *Knowl.-Based Syst.* 301, 112326 (2024) and *New J. Phys.* 23, 083020 (2021), for example.

We thank the reviewer for this helpful suggestion and for providing two valuable references. In the revised manuscript, we have expanded the Introduction to better articulate the connections

between classical game theory and reinforcement learning, especially in the context of discovering novel strategies in complex iterated environments. We now cite and discuss recent works in the field, including Xu et al., *Knowledge-Based Systems*, 2024, 301: 112326, who applied Q-learning in higher-order network settings, and Jia et al., *New Journal of Physics*, 2021, 23(8): 083020, who explored the role of local and global stimuli in reinforcement learning within social dilemmas. These studies enrich the methodological and conceptual grounding of our reinforcement learning framework (see Lines 73-86).

Pseudocode Improvement: The pseudocode section could be revised to make it more accessible to a broader audience. Adding more comments and simplifying certain steps would help readers from less technical backgrounds better understand the algorithm. This would also improve reproducibility and comprehension for researchers seeking to build upon your work.

We thank the reviewer for this thoughtful and constructive suggestion. In the revised manuscript, we have carefully revised the pseudocode in Algorithm 1 to enhance clarity and accessibility for a broader readership. Specifically, we added inline comments to annotate key steps in the algorithm (e.g., action selection, payoff computation, Q-value update), which help clarify the function of each code block. Additionally, we refined the overall structure to present the learning process in a more intuitive and streamlined manner, making it more approachable for readers who may be less familiar with reinforcement learning. These revisions aim to balance readability with the level of detail necessary for accurate implementation (see Algorithm 1).

Discussion on Noise: In the section “The dominance of MTBR in evolving populations,” the results are presented under noise-free conditions. Extending this discussion to include the effects of noise on evolutionary trajectories and the dominance of MTBR would make the analysis more comprehensive. The importance of noise for evolutionary outcomes has been studied in *New J. Phys.* 8, 22 (2006), and in many other publications in the past decades.

We thank the reviewer for this excellent suggestion and for providing a valuable literature for reference. In response, we have incorporated strategy execution noise into our simulations — specifically, modeling agents as mis-implementing their intended actions with probability η . This form of behavioral execution error is widely recognized in studies of evolutionary dynamics. We tested three levels of noise: $\eta = 0.001$, 0.01 , and 0.05 . Our results show that MTBR remains absolutely dominant under low-levels of noise (see Fig. R1 below). Even as noise increases to $\eta = 0.01$, MTBR continues to outperform all other strategies by a wide margin. When the noise level is further raised to $\eta = 0.05$, MTBR and GTFT0.3 jointly dominate the evolutionary dynamics, consistently outperforming all other strategies. These findings underscore the robustness of MTBR’s evolutionary advantage in noisy environments. We have integrated these results into the main text (Lines 332–338) and added Supplementary Fig. 4 to the Supplementary Information.

Fig. R1: **MTBR remains dominant under behavioral execution errors.** We examine the evolutionary dynamics when agents occasionally mis-execute their intended actions with probability η , introducing behavioral execution noise. Three levels of noise are considered: $\eta = 0.001$, 0.01 , and 0.05 . **a**, At low noise ($\eta = 0.001$), MTBR remains fully dominant, ultimately taking over the entire population. **b**, When $\eta = 0.01$, MTBR stabilizes at approximately 72%, coexisting with GradualTFT. **c**, When $\eta = 0.05$, GradualTFT rapidly disappears, and MTBR and GTFT0.3 become the prevailing strategies, stabilizing around 53% and 47% respectively. All other parameter settings are consistent with those used in Fig. 4 of the main text.

Terminology Consistency: The term "dominant strategies" is used frequently throughout the manuscript but is not consistently defined. It is recommended to clarify whether it refers to payoff superiority, evolutionary prevalence, or strategic stability. Providing a clear and consistent definition would avoid ambiguity and enhance the clarity of your results and conclusions.

We thank the reviewer for pointing out this confusion. In our study, we use the term "dominant strategy" to refer to a strategy that exhibits an evolutionary advantage in mixed populations — either by eventually taking over the population under evolutionary dynamics or by maintaining a high frequency in the evolutionarily stable state. To clarify this usage, we have added a definition in the Introduction (Lines 89-92), where the term is first introduced, and we have ensured that this meaning is applied consistently throughout the manuscript.

If a revision is granted, and if needed, I will be happy to review the manuscript again.

We appreciate the reviewer for the thoughtful summary of our work's significance and for the constructive suggestions, which have greatly enhanced the clarity and overall quality of the manuscript.

Replies to comments and suggestions from referee 2

Comments:

This study explores a classical topic: What constitutes a "strong" strategy in the Iterated Prisoner's Dilemma (IPD)? Using multi-agent reinforcement learning, the authors identified a strategy called memory-two bilateral reciprocity (MTBR). This strategy behaves like tit-for-tat (TFT) in most cases but cooperates when both players defect twice in a row as shown in the Supplementary Table. Using MTBR, the authors analyzed its performance in tournaments and evolutionary dynamics.

The authors found that MTBR performs well in these scenarios and claim that introducing MTBR into a population increases the overall level of cooperation.

We thank the referee for carefully reading our paper and all of the valuable suggestions below.

However, after a careful review of the manuscript, I cannot recommend it for publication. My decision is based on several critical concerns, the most significant of which is the authors' arbitrary selection of strategies (based on their subjective choice of "well-recognized strategies") as potential opponents for MTBR. This selection applies not only to the training process but also to the tournament and evolutionary dynamics. Such an approach introduces significant bias and limits the generality of the findings. For instance, MTBR would likely be exploited by the AllD strategy, which is not adequately addressed.

We thank the reviewer for raising this important point, which prompts us to clarify the generality and robustness of our findings. We address this concern from three key perspectives: First, we emphasize that nearly all strategies — including well-known ones such as Win-Stay Lose-Shift (WSLS) and Generous Tit-for-Tat (GTFT) — are to some extent exploitable by the unconditionally defecting strategy AllD in direct encounters, simply because AllD never cooperates. However, our goal in studying strategies in the Iterated Prisoner's Dilemma (IPD) is not to outperform AllD in head-to-head interactions. Rather, the focus is on identifying strategies that achieve high payoffs across diverse opponents and resist invasion in evolving populations. MTBR excels in both respects. For example, in an evolving population of MTBR and AllD, under standard IPD parameters ($R = 3, S = 0, T = 5, P = 1$), we observe the following average payoffs (over 20 rounds):

	MTBR	AllD
MTBR	2.94	0.65
AllD	2.40	1.00

While AllD exploits MTBR in direct encounters (2.40 vs. 0.65), it performs poorly against itself (1.00), compared to MTBR's performance against itself (2.94). Replicator dynamics analysis

confirms that under these conditions, ALLD cannot successfully invade a population dominated by MTBR—highlighting MTBR’s evolutionary robustness even in the face of purely defecting opponents.

Second, we agree that evaluating a strategy’s generality requires testing it against a broad and representative set of opponents. To this end, we included in the strategy pool two additional ALLD-like strategies: Hold-a-Grudge (which permanently defects after observing any defection) and Fool-Me-Once (which defects permanently after two defections from the opponent). These strategies further pressure cooperative behaviors, yet MTBR remains highly competitive in their presence. Moreover, in our additional experiments, we also explicitly included ALLD itself and found that its presence does not affect the overall dominance of MTBR (see Fig. R2).

Additionally, we clarify that the evolutionary simulations employ a significantly broader strategy pool than the one used during training. This includes not only classical strategies but also a range of Zero-Determinant (ZD) strategies. Even in this more diverse and challenging environment, MTBR consistently achieves high payoffs and effectively steers the population toward cooperation.

Finally, in line with the reviewer’s suggestion, we have expanded the strategy pool to include several recent high-performing and longer-memory strategies, such as CURE (Nature Computational Science, 2022, 2(10): 677–686), AON2 (PNAS, 2017, 114(18): 4715–4720), and Reactive-2-Partner (PNAS, 2024, 121(50): e2420125121). Even within this enriched and more competitive landscape, MTBR continues to emerge as a dominant strategy under evolutionary selection, as shown in Fig. R2 below.

In the revised manuscript, we have added a dedicated paragraph to further highlight the robustness and generality of MTBR (see Lines 450-465), along with Supplementary Fig.8 to support this conclusion. We hope these additions fully address the reviewer’s concerns.

Moreover, the authors should analyze MTBR’s properties using more rigorous and objective methods. For example, is MTBR a partner strategy for certain benefit-to-cost (b/c) ratios? Is it an evolutionarily robust strategy? These questions are fundamental for establishing the robustness of a strategy in the IPD. Established theoretical frameworks exist to address such questions, and incorporating these would greatly strengthen the research. Examples include: [1] Hilbe, Christian, Krishnendu Chatterjee, and Martin A. Nowak. "Partners and rivals in direct reciprocity." *Nature human behaviour* 2.7 (2018): 469-477. [2] Stewart, Alexander J., and Joshua B. Plotkin. "From extortion to generosity, evolution in the iterated prisoner’s dilemma." *Proceedings of the National Academy of Sciences* 110.38 (2013): 15348-15353.

We thank the reviewer for this excellent suggestion and for recommending two valuable references.

Following the reviewer’s advice, we conducted additional analyses using more rigorous methods and demonstrated that MTBR qualifies as a partner strategy in the general Iterated Prisoner’s

Fig. R2: **MTBR dominates in the presence of advanced longer-memory strategies.** We introduce Strategy Set 3, which includes three advanced strategies (namely CURE, AON2, and Reactive-2-Partner) and AllID in addition to the 15 strategies from Strategy Set 2 described in the main text. To evaluate MTBR's performance, we compare two evolving populations: one composed solely of Strategy Set 3 (a for payoff ratio $b/c = 2$, c for $b/c = 1.5$), and the other composed of Strategy Set 3 plus MTBR (b for $b/c = 2$, d for $b/c = 1.5$). Each simulation begins with 33 individuals per strategy. **a**, For $b/c = 2$, the population converges to a mixed equilibrium where GTFT0.3, GradualTFT, and CURE stabilize at 72%, 24%, and 4%, respectively. **b**, When $b/c = 2$ and MTBR is included, it ultimately dominates the entire population. **c**, For $b/c = 1.5$, GTFT0.3 dominates the population, stabilizing at approximately 97%. **d**, With $b/c = 1.5$ and MTBR present, it stabilizes at around 98% of the population. These results demonstrate that MTBR withstands competition from sophisticated, longer-memory strategies. Each trajectory represents an average over 50 independent simulation runs. Notably, CURE tends to perform relatively well in the early stages of evolution.

Dilemma, characterized by the payoff parameters R, S, T, P , under the following conditions:

$$\begin{aligned} S &< R, \\ T &< 2R - S, \\ P &< \frac{1}{2}(3R - T). \end{aligned} \tag{1}$$

In particular, for the Iterated Donation Game, i.e., $R = b - c, S = -c, T = b, P = 0$, these conditions reduce to:

$$\frac{b}{c} > \left(\frac{b}{c}\right)^* = 1.5, \tag{2}$$

where $(b/c)^*$ denotes the critical benefit-to-cost ratio. Importantly, this threshold is relatively easy to satisfy in many realistic scenarios, indicating that MTBR can function as a partner strategy across a broad range of game parameters.

Furthermore, Stewart and Plotkin have shown that in sufficiently large populations and in the absence of payoff discounting, all partner strategies are evolutionarily robust (PNAS, 2014, 111(49): 17558–17563.). Based on this result, we conclude that MTBR also qualifies as an evolutionarily robust strategy.

Below, we present the mathematical details of the rigorous proof that MTBR satisfies the conditions of a partner strategy. We consider an infinitely repeated Prisoner’s Dilemma between two players, Alice and Bob. In each round, both players independently choose to either cooperate (C) or defect (D), with the standard payoff matrix:

$$\begin{array}{c|cc} & C & D \\ \hline C & R & S \\ D & T & P \end{array} \tag{3}$$

We focus on games satisfying $T + S < 2R$, a condition that ensures mutual cooperation is socially optimal. A strategy employed by Alice is defined as a partner strategy if it meets the following two conditions (Nature Human Behaviour, 2018, 2(7): 469–477): (i) When Bob uses the same strategy as Alice, both players achieve the mutual cooperation payoff, i.e., $\pi_A = \pi_B = R$, where π_A and π_B are the long-term average payoffs for Alice and Bob, respectively; (ii) if Bob employs a different strategy, his payoff does not exceed R , and in such cases, Alice receives an equal payoff.

Before presenting the proof, we introduce two key preliminaries that underpin our analysis.

1. Memory compression. Press and Dyson (PNAS, 2012, 109(26), 10409–10413) showed that when two players have different memory lengths, the player with the longer memory can achieve the same payoff by using a shorter-memory strategy that ignores information unavailable to the opponent. This result allows us to simplify asymmetric-memory interactions into equivalent interactions where both players have the same memory length. Therefore, to evaluate how MTBR performs against arbitrary strategies, it suffices to consider its performance against opponents with memory-2 strategies.

2. Pure strategy best response. When both players use memory-2 strategies, the best response to any fixed strategy is always achieved by a pure strategy. This extends the reasoning originally developed by Press and Dyson for memory-1 strategies, and Glynatsi et al. (PNAS, 2024, 121(50): e2420125121) further generalized it to encompass memory- n reactive strategies. Specifically, the expected payoff can be written as a linear-fractional function of the opponent's strategy — meaning both the numerator and denominator are linear in the opponent's strategy variables. Since the strategy space is a compact, convex hypercube $[0, 1]^{16}$, the maximum is always attained at one of its corners, i.e., a pure strategy. A sketch proof is provided in Supplementary Note 3.

With these two preliminaries in place, we only need to show that MTBR satisfies the partner conditions when interacting with any of the 2^{16} possible pure memory-2 strategies.

The interaction state between two players adopting memory-2 strategy is represented as

$$((a_1^{t-2}, a_2^{t-2}), (a_1^{t-1}, a_2^{t-1})), \quad (4)$$

where $a_1^{t-1} = 1$ (or 0) indicates player 1 cooperated (or defected) last round and $a_1^{t-2} = 1$ (or 0) indicates player 1 cooperated (or defected) two rounds ago. The same convention applies to player 2. To simplify analysis, we map each joint action history to a unique scalar index using the rule:

$$i = 15 - (2^0 \cdot a_2^{t-1} + 2^1 \cdot a_1^{t-1} + 2^2 \cdot a_2^{t-2} + 2^3 \cdot a_1^{t-2}). \quad (5)$$

For example, if both players cooperated in each of the last two rounds, the mapping yields:

$$i = 15 - (2^0 + 2^1 + 2^2 + 2^3) = 15 - 15 = 0. \quad (6)$$

This indexing ensures that lower values of i correspond to more cooperative histories, with state 0 representing full cooperation over the last two rounds, and state 15 corresponding to mutual defection. A complete mapping between joint histories and their respective indices is provided in Table 1.

We define a memory-2 strategy as a 16-dimensional vector:

$$\mathbf{p} = (p_0, p_1, p_2, \dots, p_{15}), \quad (7)$$

where p_i denotes the probability of cooperating when the system is in state i , as defined by the encoded history of actions over the previous two rounds. Specifically, the MTBR (Mark the Best Response) strategy, when adopted by player 1, is represented as:

$$\mathbf{p}_{\text{MTBR}} = (1, 0, 1, 0, 1, 0, 1, 0, 1, 0, 1, 0, 1, 0, 1, 1). \quad (8)$$

This strategy prescribes deterministic actions in each state, alternating between cooperation and defection, with full cooperation in the initial state ($i = 0$) and mutual defection state ($i = 15$). Let M denote the state transition matrix that governs the Markov process arising from the interaction between MTBR (as player 1) and a given memory-2 pure strategy. Moreover, we

a_1^{t-2}	a_2^{t-2}	a_1^{t-1}	a_2^{t-1}	State index
1	1	1	1	0
1	1	1	0	1
1	1	0	1	2
1	1	0	0	3
1	0	1	1	4
1	0	1	0	5
1	0	0	1	6
1	0	0	0	7
0	1	1	1	8
0	1	1	0	9
0	1	0	1	10
0	1	0	0	11
0	0	1	1	12
0	0	1	0	13
0	0	0	1	14
0	0	0	0	15

Table 1: **Relationship between interaction state and index.**

denote $v_i(t)$ as the probability that the joint actions of the two players in the past two rounds correspond to the state i . Then the stationary (invariant) distribution \mathbf{v} over the 16 interaction states can be obtained by solving

$$\begin{aligned}\mathbf{v}^T M &= \mathbf{v}^T, \\ \mathbf{v}^T \mathbf{1} &= 1.\end{aligned}\tag{9}$$

Once \mathbf{v} is known, the expected payoffs for both players can be computed. The expected payoff for MTBR (player 1) is

$$\pi_1 = \mathbf{v}^T \cdot \mathbf{g}_1,\tag{10}$$

and the expected payoff for the opponent (player 2) is

$$\pi_2 = \mathbf{v}^T \cdot \mathbf{g}_2,\tag{11}$$

where the payoff vectors are defined as follows:

$$\mathbf{g}_1 = (R, S, T, P, R, S, T, P, R, S, T, P, R, S, T, P),\tag{12}$$

$$\mathbf{g}_2 = (R, T, S, P, R, T, S, P, R, T, S, P, R, T, S, P).\tag{13}$$

To determine whether MTBR qualifies as a partner strategy, we compare π_2 with the mutual cooperation payoff R . MTBR is a partner strategy if it never yields the opponent a payoff

greater than R when facing any pure memory-2 strategy. Through computation, we find that this condition is equivalent to verifying the following inequalities:

$$\begin{aligned}
\frac{2P}{3} + \frac{T}{3} &< R, \\
\frac{2P}{5} + \frac{2R}{5} + \frac{T}{5} &< R, \\
\frac{2P}{5} + \frac{S}{5} + \frac{2T}{5} &< R, \\
\frac{2P}{7} + \frac{2R}{7} + \frac{S}{7} + \frac{2T}{7} &< R, \\
\frac{P}{2} + \frac{R}{4} + \frac{T}{4} &< R, \\
\frac{P}{3} + \frac{R}{6} + \frac{S}{6} + \frac{T}{3} &< R, \\
\frac{P}{3} + \frac{S}{3} + \frac{T}{3} &< R, \\
\frac{P}{4} + \frac{R}{4} + \frac{S}{4} + \frac{T}{4} &< R, \\
\frac{P}{5} + \frac{2R}{5} + \frac{S}{5} + \frac{T}{5} &< R, \\
\frac{P}{6} + \frac{R}{6} + \frac{S}{3} + \frac{T}{3} &< R, \\
\frac{P}{7} + \frac{2R}{7} + \frac{2S}{7} + \frac{2T}{7} &< R, \\
\frac{R}{2} + \frac{S}{4} + \frac{T}{4} &< R, \\
\frac{R}{3} + \frac{S}{3} + \frac{T}{3} &< R, \\
\frac{S}{2} + \frac{T}{2} &< R.
\end{aligned} \tag{14}$$

By simplifying the inequalities above, we obtain the following necessary and sufficient conditions:

$$\begin{aligned}
S &< R, \\
T &< 2R - S, \\
P &< \frac{1}{2}(3R - T).
\end{aligned} \tag{15}$$

In other words, MTBR qualifies as a partner strategy in the Iterated Prisoner's Dilemma (IPD) if all the above conditions are satisfied.

Under the standard donation game payoff structure,

$$R = b - c, \quad S = -c, \quad T = b, \quad P = 0, \tag{16}$$

the above inequalities reduce to a single threshold condition:

$$\frac{b}{c} > \left(\frac{b}{c}\right)^* = 1.5. \tag{17}$$

Hence, MTBR qualifies as a partner strategy in the donation game if and only if the benefit-to-cost ratio exceeds 1.5.

In the revised manuscript, we have cited the two references, added a paragraph to discuss the partner studies (see Lines 467-482), and also supplemented the mathematical details to the Supplementary Information. We have also made the invariant distribution data and corresponding code used in the analysis publicly available on GitHub.

Additionally, the manuscript overlooks a substantial body of prior research on longer-memory strategies. While many studies focus on memory-one strategies, there is also extensive work on longer-memory strategies, which the authors fail to acknowledge. A meaningful comparison with this literature is essential to contextualize the findings. For instance: - Hilbe, Christian, et al. "Memory-n strategies of direct reciprocity." *PNAS* 114.18 (2017): 4715-4720. - Do Yi, Su, Seung Ki Baek, and Jung-Kyoo Choi. "Combination with anti-tit-for-tat remedies problems of tit-for-tat." *J. Theor. Biol.* 412 (2017): 1-7. - Murase, Yohsuke, and Seung Ki Baek. "Five rules for friendly rivalry in direct reciprocity." *Scientific Reports* 10.1 (2020): 16904. - Li, Juan, et al. "Evolution of cooperation through cumulative reciprocity." *Nat. Comput. Sci.* 2.10 (2022): 677-686. - Murase, Yohsuke, and Seung Ki Baek. "Grouping promotes both partnership and rivalry with long memory in direct reciprocity." *PLOS Comput. Biol.* 19.6 (2023): e1011228. - Glynatsi, Nikoleta E., et al. "Conditional cooperation with longer memory." *PNAS* 121.50 (2024): e2420125121.

We thank the reviewer for this insightful comment and for suggesting five valuable references. In response, we have carefully reviewed recent advances on longer-memory strategies and incorporated a focused discussion into the revised manuscript. In particular, we examine three representative strategies — CURE, AON2, and Reactive-2-Partner — each representing a distinct mechanism for supporting cooperation in iterated games.

The CURE strategy (Li et al., *Nature Computational Science*, 2022, 2(10): 677–686) promotes cooperation through a mechanism of cumulative reciprocity: players accumulate cooperation credits and respond to opponents based on long-term cooperation history. This design allows CURE to tolerate occasional defections and reward consistently cooperative opponents. As a result, CURE performs particularly well against strict retaliatory strategies such as Hold-a-Grudge. However, its fixed leniency also introduces vulnerabilities: in finite-horizon settings like ours (20 rounds), delayed retaliation can be exploited by opportunistic strategies that defect in the early or final rounds. In our simulations, CURE exhibits strong performance early in the evolutionary process but is eventually outcompeted by MTBR, which more efficiently balances cooperation and timely punishment in fixed-length interactions.

The AON2 strategy (Hilbe et al., *PNAS*, 2017, 114(18): 4715–4720) represents a class of memory-2 all-or-none strategies. AON2 players cooperate only when both players' last two moves are identical, enforcing strict symmetry in interaction. This rule makes AON2 highly effective at maintaining cooperation when coordination is achieved, but even minor deviations or asynchronous behavior can trigger persistent defection. Moreover, AON2's rule-based rigidity con-

trasts with MTBR’s learned flexibility: MTBR can recover from temporary asymmetries and adapt to diverse opponents, which gives it a significant advantage in evolutionary settings with multiple competing strategies.

The Reactive-2-Partner strategy (Glynatsi et al., PNAS, 2024, 121(50): e2420125121) is analytically constructed to be a partner strategy within the space of reactive memory-2 strategies. It is not noise-dependent, but performs best under three specific conditions: long memory, low cost-to-benefit ratio, and infinitely repeated interactions. Under these theoretical assumptions, Reactive-2-Partner is shown to be both individually rational and stable in the sense of Nash equilibrium. However, its performance is highly context-sensitive: in finite-horizon games such as ours, where interactions last only 20 rounds and there is no uncertainty about when the game ends, the assumptions underlying its theoretical advantage no longer hold. Our work instead focuses on adaptive success in short-term interactions, where strategies must not only be cooperative but also rapidly responsive. In such scenarios, MTBR demonstrates higher evolutionary fitness.

In our evolutionary simulations (Supplementary Fig. 8), MTBR consistently outperforms these longer-memory strategies. It fully dominates the population under $b/c = 2$, and remains dominant at 98% under $b/c = 1.5$. Notably, CURE achieves competitive payoffs early in the simulations due to its forgiving design, but fails to maintain long-term prevalence. These findings underscore the robustness and adaptability of MTBR when compared to theoretically elegant but structurally constrained longer-memory strategies.

Yi *et al.* (Journal of Theoretical Biology, 2017, 412: 1–7) introduced TFT-ATFT, a hybrid strategy designed to recover from erroneous defections in noisy environments. Their analysis focuses on stochastic tournaments with implementation errors and shows how carefully constructed response sequences can mitigate short-term misunderstandings. However, the strategy’s advantage is most pronounced in environments where such noise is prevalent, which differs from our setting of deterministic, finite-length interactions.

Murase and Baek (Scientific Reports, 2020, 10: 16904) formalized the concept of *friendly rival* strategies — those that simultaneously achieve mutual cooperation and prevent exploitation. Through exhaustive enumeration, they identified the CAPRI strategy, which satisfies key axioms of efficiency, defensibility, and distinguishability. Their work stands out for the theoretical rigor and clarity in defining long-memory cooperation, but again assumes infinite repetition and non-zero noise levels.

More recently, Murase and Baek (PLOS Computational Biology, 2023, 19(6): e1011228) demonstrated that friendly rival strategies emerge and dominate only when memory length is sufficiently long and the population has internal group structure. Their simulations show that in group-structured populations, long-memory strategies like CAPRI can outperform both partners and rivals even under adverse conditions such as low benefit-to-cost ratios. This work highlights the essential interaction between memory and population structure — an important complement to our own study focused on well-mixed populations and finite repeated games.

Taken together, these contributions offer valuable insights into the design and emergence of long-memory strategies in the presence of noise and structural complexity. Our study complements

this body of work by focusing on deterministic interactions without implementation errors—a setting in which we explore whether reinforcement-trained memory-two strategies, such as MTBR, can achieve evolutionary dominance within a diverse and competitive strategic landscape. In the revised manuscript, we have cited these references and added a paragraph to discuss longer memory strategies. (see Lines 514-528)

Furthermore, the advantage of MTBR over other strategies appears marginal. As shown in Figure 3, its performance is comparable to GTFT. Similarly, the claim in Figure 4c that introducing MTBR improves the overall cooperation level is not compelling, as the improvement is only 0.03. The visual presentation exaggerates the significance of this result.

This is an excellent point, and we address this concern from three aspects:

First, GTFT is widely recognized as a successful strategy capable of coordinating populations toward high levels of cooperation. As shown in Fig. 4c, when GTFT dominates, it drives the population to an average payoff of 2.90 — approaching the mutual cooperation payoff of 3.00. Even so, MTBR is able to further elevate cooperation, resulting in an average payoff of 2.93. While this is only a 0.03 increase, it is noteworthy given the narrow margin between 2.90 and the theoretical maximum of 3.00. Considering the stochastic nature of initial interactions, this gain — amounting to 30% of the remaining possible improvement — is substantial. More importantly, the population composition shifts dramatically: the introduction of MTBR transforms a coexistence of multiple strategies into a near-monoculture dominated by MTBR, leading to a qualitatively different evolutionary outcome.

Second, the cooperation-promoting effect of MTBR becomes even more pronounced in settings where GTFT loses its dominance. For instance, in a small population of approximately 50 individuals (Fig. R3a), the absence of GTFT0.3 results in a low average payoff of 1.78. Introducing GTFT0.3 increases the average payoff to 2.44, indicating partial cooperation. However, introducing MTBR instead leads to a much higher average payoff of 2.94 — nearly reaching the full cooperation payoff. This demonstrates that MTBR can outperform GTFT0.3 even in more challenging evolutionary settings. Furthermore, when both MTBR and GTFT0.3 are introduced, the resulting dynamics again yield a high average payoff, underscoring MTBR’s robust cooperation-promoting capabilities.

In the revised manuscript, we have added a paragraph and an accompanying Supplementary Fig. 3 to further elaborate on these findings and highlight the cooperation-enhancing effects of MTBR. (see Lines 300-311)

Overall, we have clarified our statements and conducted several new analyses. We believe these revisions and additional results effectively address the reviewer’s concerns.

Fig. R3: **MTBR promotes cooperation in a small and competitive strategy set.** We define *Strategy Set 4* consisting of nine strategies: TFT, WSLs, Hold-a-Grudge, ZDExtort2, ZDExtort2v2, ZDExtort3, ZDExtort4, ZDMischief, and ALLD. Each strategy is initialized with 5 individuals. Given the relatively small population sizes (ranging from 45 to 55), the evolutionary dynamics tend to converge to monomorphic populations, where a single strategy ultimately dominates. Therefore, each trajectory represents the probability of a strategy fully taking over the population, averaged over 50 independent runs. **a**, Evolutionary outcome under Strategy Set 4 alone. Hold-a-Grudge dominates with 90% probability, followed by WSLs (8%) and ALLD (2%). The average payoff at equilibrium is 1.78. **b**, Evolution with Strategy Set 4 plus GTFT0.3. The population stabilizes under GTFT0.3 (30%), WSLs (36%), Hold-a-Grudge (32%), and ALLD (2%). The average population payoff increases to 2.44. **c**, Evolution with Strategy Set 4 plus MTBR. MTBR completely takes over the population in all runs (100% dominance), yielding an average payoff of 2.94. **d**, Evolution with Strategy Set 4 plus both GTFT0.3 and MTBR. MTBR dominates in 96% of runs, and GTFT0.3 in 4%. The average population payoff remains high at 2.93. These results show that while GTFT0.3 can moderately promote cooperation in this difficult setting, MTBR consistently achieves near-optimal outcomes by fully stabilizing cooperation, even when coexisting with other cooperative strategies. This underscores MTBR's robustness and superior cooperation-promoting capability under constrained evolutionary conditions.

Replies to comments and suggestions from referee 3

Comments:

In this study, the authors investigate dominant strategies in repeated games using a reinforcement learning framework. Their work provides a novel approach to analyzing dominant strategies and reports an intriguing new strategy—Memory-Two Bilateral Reciprocity (MTBR). The authors demonstrate that MTBR stands out among various strategies, facilitating mutual cooperation and enhancing overall payoffs. Moreover, they employ an evolutionary model to study the evolutionary stability of this strategy. I appreciate the introduction of this innovative framework and the discovery of a new, special strategy, which contribute to advancing research in evolutionary game theory and cooperation dynamics. This is very interesting and meaningful work; however, before I agree to accept it, the authors could benefit from clarifying certain aspects. Below are my comments:

We sincerely appreciate the reviewer’s detailed reading and summary of our manuscript, and all the constructive suggestions below.

The authors could benefit from providing a more detailed explanation of the design of agents’ action objectives. Within the reinforcement learning framework, an agent’s actions are fundamentally guided by the goal of maximizing an objective or utility function (i.e., here W_{p_x}), which the authors consider as the relative advantage over opponents and their own payoffs. I would appreciate further clarification on the motivation and rationale behind this design of W_{p_x} .

We thank the reviewer for encouraging us to clarify the design of agents’ action objectives. In our reinforcement learning framework, the agent’s objective function W_{p_x} is crafted to capture two key aspects of evolutionary success: (i) achieving a relative advantage — i.e., obtaining higher payoffs than the opponent — and (ii) securing high absolute payoffs for themselves. These objectives reflect distinct evolutionary pressures.

The relative advantage ensures that the learned strategy can outperform opponent strategies in direct encounters, thereby avoiding exploitation. However, a strategy focused solely on relative gain may lead to mutual defection, resulting in low payoffs for both parties. In an evolving population, such strategies risk being outcompeted by those that generate higher absolute payoffs through cooperation.

Therefore, maintaining high absolute payoffs is equally critical, as it indicates a capacity for fostering coordination and mutual cooperation — traits that enhance a strategy’s viability within the population. This dual-objective design enables the discovery of strategies that are both competitively robust and socially beneficial. We have incorporated this explanation into the main text (Lines 158–165) to clarify the rationale behind the construction of W_{p_x} .

How are the initial actions of the RL-based agents determined? Do they randomly choose between cooperation and defection? Additionally, does the choice of initial action affect the results? For example, certain strategies, such as the Grim strategy, are particularly sensitive to initial actions in interactions. Specifically, after each training step (L steps), when individuals are rematched to play the game, how is the initial strategy defined in this context?

We thank the reviewer for prompting us to clarify the training details. In our reinforcement learning framework, each agent maintains a Q-table that estimates the expected long-term payoff for each action in each state. At the start of training, all Q-values are initialized to zero. Consequently, in the first round of the initial game, agents have no prior preference and randomly choose between cooperation and defection, as the reviewer noted.

This random action selection also applies when agents are rematched to play new repeated games. This design reflects two considerations: (i) in the first round of each new game, there is no interaction history between agents and their new opponents, so agents cannot infer any information and must choose randomly; (ii) randomizing the first move exposes agents to a wider range of initial scenarios, helping them learn strategies that are effective across diverse contexts. We have also examined alternative initialization schemes, such as always cooperating or always defecting in the first round. For instance, if agents always start with cooperation, classic strategies like TFT, GTFT0.3, WSLS, and GradualTFT may converge to stable mutual cooperation in noise-free settings, making their long-term behaviors difficult to distinguish. Moreover, agents may fail to learn how to defend against exploitation, and thus become vulnerable to defecting strategies such as ALLD. Conversely, if the initial move is always defection, unforgiving strategies such as Grim (referred to as Hold-a-Grudge in our manuscript) may trigger permanent retaliation, preventing the emergence of cooperation and often leading the agent to converge to uncooperative strategies like ALLD.

As the reviewer rightly pointed out, the initial action can influence the strategies that emerge. Our choice of random initial actions ensures a more robust and meaningful evaluation of strategic responses during training, which ultimately led to the discovery of MTBR.

We thank the reviewer again for this insightful question. We have clarified this point in the revised manuscript. (Lines 147-152)

During the training process, does the probability of an agent encountering a mentor (which is equivalent to the number of mentors) lead to different training outcomes? Additionally, does it affect the convergence of the RL training?

Good point. In our framework, agents are stochastically matched with either mentors or other agents. By default, we use an equal number of agents and mentors, and each agent is matched with a mentor or another agent with equal probability.

We have also explored alternative configurations—for example, matching each agent with a mentor with probability 25% (and with another agent with probability 75%), or vice versa. These variations in mentor encounter probability lead to different training dynamics. Specifically, when

agents are more likely to interact with mentors, MTBR tends to emerge more frequently across multiple training runs.

Importantly, even when the probability of encountering a mentor is lower, MTBR still occasionally appears. This suggests that the multi-agent learning can substantially narrow the effective strategy search space compared to purely random exploration, enabling the efficient discovery of functional strategies under specific configurations.

Moreover, while varying the mentor encounter probability influences the types of strategies that emerge, it does not affect the convergence of the reinforcement learning process in our setting. Specifically, we train 49 agents independently in each experiment, and each agent maintains its own Q-table. As long as the training runs for a sufficient number of steps, the Q-table of each agent consistently converges to a stable form. In this sense, the learning process is robust across different mentor configurations, and convergence is observed at the individual strategy level for all agents.

We have clarified this point in the revised manuscript (Lines 419-433).

I noticed that the authors mention, “Our goal is to explore strategies that not only dominate the opponent but also coordinate for a high collective payoff.” Does this imply an exploration of prosocial or other-regarding strategies? Given that recent studies have discussed and emphasized human prosocial preferences [e.g., PNAS 2024, 121(49): e2412195121], could the authors provide some discussion on this aspect?

We thank the reviewer for this excellent point and for providing a valuable reference. Indeed, our work addresses the emergence of prosocial behavior in strategic settings, but through a reinforcement learning lens rather than by assuming innate prosocial preferences. Our goal is to discover strategies that both outperform opponents and sustain high payoffs, without explicitly constraining the agents to behave prosocially or exhibit other-regarding preferences (see the recent human-centered study (PNAS, 2024, 121(49): e2412195121)).

Interestingly, the strategy that emerges from our framework, Memory-Two Bilateral Reciprocity (MTBR), embodies a form of contingent cooperation: it actively fosters cooperative interactions with cooperative partners while effectively resisting exploitation. This is consistent with the spirit of classical strategies like Tit-for-Tat, Generous-TFT, and GradualTFT, which aim not to exploit others, yet are resilient against being exploited themselves.

While our setup does not impose any prosocial bias, MTBR behaves in a prosocial or other-regarding manner by promoting mutual cooperation and facilitating coordination toward higher collective payoffs. More importantly, our findings suggest that prosocial-like strategies can arise naturally from reinforcement learning dynamics, shaped by environmental feedback and structural incentives rather than predefined preferences — an insight that opens up exciting avenues for future research.

We have incorporated this reference and added a discussion of these points in the revised manuscript (see Lines 557-573).

The specific definition of the state in the model is unclear. Could the authors further clarify what the elements in the state set represent? Stating this in the main text could help readers intuitively understand the concept without relying on supplementary materials. An example might be useful to illustrate this point.

We thank the reviewer for this helpful suggestion. In our model, the *state* refers to the information available to an agent at the time of decision-making. Specifically, for memory-two strategies, the state is defined by the action pairs taken by the agent and its opponent in the two preceding rounds. The state is represented as a four-element tuple: (1) the opponent’s action two rounds ago, (2) the agent’s action two rounds ago, (3) the opponent’s action last round, and (4) the agent’s action last round. This design enables agents to condition their current decisions on the complete two-round interaction history.

We have now incorporated this clarification into the main text (Lines 125–132), along with a concrete example to help readers intuitively understand how states are defined in our framework.

I noticed that the authors use a specific payoff matrix. I would like to understand whether the results are dependent on a specific particular matrix. Recent studies have introduced the concept of dilemma strength in the Prisoner’s Dilemma [e.g., *Physics of life reviews*, 2015, 14: 1-30 (2015)], providing a quantifiable standard across the payoff matrix. This allows research on the Prisoner’s Dilemma to be independent of specific payoff matrices. Could the authors include related results or discussions on this? This might enhance the generalizability of the findings.

We thank the reviewer for this thoughtful suggestion. In fact, in the section titled “The dominance of MTBR in various games and networks,” we already adopt the same classification scheme of social dilemmas as proposed in the referenced work (*Physics of Life Reviews*, 2015, 14: 1–30). This includes the Prisoner’s Dilemma, Snowdrift Game (also known as the Hawk–Dove Game), Stag-Hunt Game, and Harmony Game. These games correspond to distinct regions in the $(T - R, S - P)$ payoff space, capturing varying degrees of dilemma strength. We have now explicitly cited this reference in the main text (Lines 344–347) to clarify this connection and underscore the generalizability of our findings.

In the results section, the authors mention the “prosperous state.” Could the authors clarify what this refers to and explain the criteria used to determine it?

We thank the review for pointing out this confusion. In our work, the term “prosperous state” refers to a situation in which the average payoff across the entire population is high. Specifically, we observe that the introduction of the MTBR strategy consistently raises the global average payoff in the population. Based on this observation, we argue that MTBR effectively guides the system toward a more prosperous state. We have clarified this terminology in the revised manuscript (see Lines 246-248).

In “The dominance of MTBR in evolving populations”, the authors combine the MTBR strategy they discovered with an evolutionary model to derive results. I would like to understand how the MTBR strategy is defined within this model. Does it refer to individuals using a pre-trained q-table for decision-making interactions?

Yes, exactly. In our evolutionary simulations, the MTBR strategy refers to agents that make decisions based on a specific, pre-trained Q-table obtained through our multi-agent reinforcement learning framework. This Q-table defines the agent’s action choices in response to each state, and remains fixed throughout the evolutionary process. We have provided a complete representation of MTBR’s behavior in Supplementary Table 1, which lists the action selected for each possible state. In addition, when there is no historical information (i.e., the first round of an interaction), MTBR selects its initial move randomly between cooperation and defection.

Additionally, does the model account for mutations? Some literature has also emphasized the importance of mutations in evolutionary models [e.g., PNAS, 2023, 120(20): e2221080120.].

Good question. We have extended our analysis by incorporating strategy mutations into the evolutionary models, following the approach used by Tkadlec et al. (PNAS, 2023, 120(20): e2221080120). Specifically, we examine three mutation rates: $\mu = 0.001$, $\mu = 0.01$, and $\mu = 0.1$. Our results show that MTBR remains dominant across all mutation rates (see Fig. R4 below), demonstrating its robustness even under significant stochastic perturbations. These findings align with the study cited by the reviewer and further support the resilience and evolutionary advantage of MTBR.

We have incorporated these results into the revised manuscript (see Lines 435-448) and added a new Supplementary Fig. 7 to illustrate the outcomes.

Fig. R4: **MTBR dominates the evolving population in the presence of mutations.** We examined the impact of three representative mutation rates: $\mu = 0.001$, 0.01 , and 0.1 . **a**, For $\mu = 0.001$, the evolutionary dynamics closely resemble those of the no-mutation case, with MTBR rapidly dominating the population. **b**, At $\mu = 0.01$, MTBR remains dominant, though the convergence speed is slightly reduced. **c**, At $\mu = 0.1$, persistent strategic diversity emerges: MTBR occupies 77% of the population, while GradualTFT and GTFT0.3 maintain stable proportions of 9% and 2%, respectively. This outcome is consistent with theoretical expectations that higher mutation rates prevent fixation and promote the coexistence of multiple cooperative strategies.

Replies to comments and suggestions from referee 4

Comments:

In premising repeated game situations, the authors found a new strategy, named memory-two bilateral reciprocity (MTBR), using a Q-learning framework. This strategy would not be easily detected by human intuition. This approach is more creative than the usual reinforcement learning applications. The authors explored simulations to show that MTBR drives up payoffs of well-known strategies such as TFT, PAVLOV, and ZD strategies (Fig. 3), which is persuasive. Moreover, they examined MTBR in symmetric 2x2 games (PD, Chicken, Stag Hunt, and Trivial games). MTBR loses dominance in the region of $S + T > 2R$. This is important. The authors mentioned ST-reciprocity and R-reciprocity, and it may be that longer memory is needed to support ST-reciprocity. Please expand this discussion and cite: (i) PLOS One 8 (8), e71961, 2013 (ii) BioSystems 90: 728–737.

We thank the reviewer for this insightful comment and for providing two valuable references. These works offer important theoretical foundations for understanding reciprocal behavior in games such as the Chicken game, where alternating cooperation and defection may lead to higher payoffs than mutual cooperation.

In our current study, the MTBR strategy emerged from training within the repeated Prisoner's Dilemma environment, where mutual cooperation typically maximizes payoffs (i.e., R-reciprocity). MTBR bases its decisions solely on the last two rounds of mutual actions (cooperation or defection) and does not have access to actual payoff values during gameplay. Consequently, it is unable to detect whether the condition $S + T > 2R$ holds, and is therefore not designed to support ST-reciprocity. We agree that enabling ST-reciprocity likely requires agents with longer memory and/or representations that incorporate payoff-based information. Future research could explore this possibility by training agents directly in Snowdrift-like environments or by enriching their state representations with recent payoff histories.

In the revised manuscript, we have cited the two references and added a brief discussion highlighting this promising direction for identifying dominant strategies that support ST-reciprocity (see Lines 385-398).

As a whole, I have a positive feeling on the MS...

We thank the reviewer for the thorough review and insightful suggestions.

Replies to comments and suggestions from referee 1

Comments:

The authors have revised their manuscript comprehensively and with love to detail. I warmly recommend publication in present form.

We sincerely thank Reviewer #1 for the very positive and encouraging evaluation of our revised manuscript. We are grateful for the reviewer's recognition of our efforts and are pleased that they find the current version suitable for publication.

Replies to comments and suggestions from referee 2

Comments:

While I appreciate the substantial effort that the authors have made to improve the manuscript, I don't recommend publication of this manuscript since the critical issues remain.

First and foremost, as I mentioned in my previous comment, the authors evaluated the strategy against a specific choice of the strategy set. While they introduced another set of strategies that indeed partially addresses the issue, the concern I raised remains unresolved.

I'm not criticizing the issue in MTBR itself, but rather the methodology employed by the authors in assessing the strategy. Every strategy has its own strengths and weaknesses, and the authors failed to present these objectively. For instance, they should have fairly compared the strategy with strategies in a given complexity class, such as memory-1 or memory-2 mixed strategies.

For example, if the population predominantly consists of ALLD, MTBR would likely fail miserably. Again, I'm not suggesting that the strategy needs to be robust against ALLD (which is obviously impossible). Instead, I'm emphasizing that the success of a strategy depends on the specific environmental setting we assumed. We need to demonstrate when the strategy performs well and when it fails.

We thank the referee for suggesting an evaluation of MTBR's performance within a broader strategy class. To address this, we conducted additional analyses. Specifically, in the evolutionary system consisting of the original eight focal strategies (MTBR, Hold-a-Grudge, Fool-me-Once, OmegaTFT, GradualTFT, CURE, AON2, and Reactive-2), we introduced ALLD and 11^4 additional memory-one strategies. A memory-one strategy is denoted by $(p_{CC}, p_{CD}, p_{DC}, p_{DD})$, where each $p_{XY} \in [0, 1]$ represents the probability of cooperating after the focal player chose X and the opponent chose Y in the previous round. We sampled 11^4 such strategies by drawing each p_{XY} uniformly from $[0, 1]$. The initial probability of cooperation was set to 0.5 (we also examined other initial conditions, and the results were qualitatively consistent). The simulation setup followed Fig. 3 of *Nature Computational Science* (2022, 2(10): 677–686): mutations occur every 2,000 generations at a rate of 10%; during each mutation, all existing strategies reduce their abundance to 99.9%, and a new strategy is introduced at 0.1%.

Our results show that MTBR becomes the sole surviving strategy in approximately 70% of independent runs, while ALLD dominates in the remaining 30%. Importantly, the 70% figure refers to MTBR taking over the entire population, not to a stationary frequency of 70%. We interpret these outcomes as follows: in noise-free conditions, exploitative strategies such as ALLD can spread rapidly at the beginning, since the initial population includes many highly cooperative strategies (akin to ALLC) that are easily exploited. Once ALLD has taken over, MTBR cannot invade because it too is exploited by ALLD. By contrast, MTBR can only establish itself when cooperative strategies survive long enough for its invasion-resistant and cooperative properties to take effect, ultimately driving the population toward a cooperative state. Notably, when-

ever MTBR achieved dominance, we observed no successful invasions by alternative memory-one strategies, highlighting the robustness of MTBR once established.

In the revised manuscript, we have included these evolutionary outcomes of MTBR within the memory-one strategy class. We also acknowledge the importance of examining strategy performance under noisy conditions. Our analysis shows that MTBR remains robust under low to moderate levels of execution error, where its cooperative advantage is well preserved. As the error rate increases further, the effectiveness of MTBR gradually declines, which is consistent with the general behavior of many cooperation-promoting strategies in highly noisy environments. Introducing noise is itself an important extension that requires substantial additional work, and we plan to investigate dominant strategies in noisy environments in a separate follow-up study. We have clarified this scope of applicability in the revised manuscript so that the assumptions and limitations of our work are made transparent.

Comments:

In the revised manuscript, the authors presented a more robust analysis to determine whether the strategy is a partner strategy or not and to assess its evolutionary robustness. I commend this direction. However, despite their claim, MTBR is not a partner strategy. I calculated the stationary distribution using Eq.(8) in the rebuttal letter and found that the strategy fails to meet the first condition of being a partner: it is not fully self-cooperative under infinitesimal error.

We thank the reviewer for carefully examining the condition of a partner strategy. However, the analysis is based on an incorrect premise: it evaluates the self-cooperation condition under infinitesimal error, whereas the definition of a partner strategy is explicitly grounded in the error-free repeated game.

The formal definition of a partner strategy is given in Nature Human Behaviour (2018, 2(7): 469–477):

When Alice and Bob play a repeated PD with $T + S < 2R$, Alice applies a ‘partner strategy’ (called ‘good strategy’ by Akin) if the following two conditions hold: (1) If Bob applies the same strategy as Alice, both get the mutual cooperation payoff, $\pi_A = \pi_B = R$; (2) By applying a different strategy, Bob can get at most R , in which case Alice gets the same payoff. That is, if $\pi_B \geq R$ then $\pi_A = \pi_B = R$.

Based on these two conditions, it is clear that the classification of a strategy as a partner strategy is independent of its performance in the presence of errors. Moreover, Hilbe et al. (Nature Human Behaviour, 2018, 2(7): 469–477) emphasize that

Whether a given strategy qualifies as a partner or rival depends on the payoff values and the continuation probability. For high continuation probabilities and a considerable benefit to cooperation, the set of partner strategies includes TFT, GTFT, WSLS and Grim.

Fig. R1: **Evolutionary dynamics of MTBR within a memory-1 strategy class.** We examine an evolutionary system that includes the original eight focal strategies (MTBR, Hold-a-Grudge, Fool-me-Once, OmegaTFT, GradualTFT, CURE, AON2, and Reactive-2), together with ALLD and 11^4 additional memory-one strategies. Initially, all strategies are introduced at equal abundance. In the long run, MTBR becomes the sole surviving strategy in approximately 70% of independent runs, while ALLD dominates in the remaining 30%. For clarity, we present only those strategies that reach a frequency of at least 10% at some point during the evolutionary process. Each curve reports the average frequency over 100 independent runs.

Two Steps Ago (Opponent)	Two Steps Ago (Self)	One Step Ago (Opponent)	One Step Ago (Self)	Strategy Choice
Cooperate	Cooperate	Cooperate	Cooperate	Cooperate
Cooperate	Cooperate	Cooperate	Defect	Cooperate
Cooperate	Cooperate	Defect	Cooperate	Defect
Cooperate	Cooperate	Defect	Defect	Defect
Cooperate	Defect	Cooperate	Cooperate	Cooperate
Cooperate	Defect	Cooperate	Defect	Cooperate
Cooperate	Defect	Defect	Cooperate	Defect
Cooperate	Defect	Defect	Defect	Defect
Defect	Cooperate	Cooperate	Cooperate	Cooperate
Defect	Cooperate	Cooperate	Defect	Cooperate
Defect	Cooperate	Defect	Cooperate	Defect
Defect	Cooperate	Defect	Defect	Defect
Defect	Defect	Cooperate	Cooperate	Cooperate
Defect	Defect	Cooperate	Defect	Cooperate
Defect	Defect	Defect	Cooperate	Defect
Defect	Defect	Defect	Defect	Cooperate
-	-	Cooperate	Cooperate	Cooperate
-	-	Cooperate	Defect	Cooperate
-	-	Defect	Cooperate	Cooperate
-	-	Defect	Defect	Defect

Table 1: **The detailed formulation of MTBR.**

This further underscores that the definition is meaningful only under error-free conditions. If trembles (infinitesimal errors) are introduced, both TFT and Grim admit non-cooperative long-run outcomes in self-play and fail to sustain self-cooperation, contradicting the claim that TFT and Grim qualify as partner strategies.

Under error-free conditions, the interaction between two MTBR strategies depends on the states reached in the first two rounds. As shown in Supplementary Table 1 and Table 1 here, the action of MTBR in round 2 ensures that the system falls into one of the states 0, 3, 4, 7, 8, 11, 12, 15 (see Supplementary Table 2 and Table 2 here for details). From any state in this subset, the play converges to the fully cooperative state (state 1). Consequently, both players achieve $\pi_A = \pi_B = R$, and condition (1) is satisfied.

The reviewer's assertion that MTBR fails the first condition of a partner strategy is therefore incorrect. While a non-cooperative stationary distribution exists in theory, it is not reachable

a_1^{t-2}	a_2^{t-2}	a_1^{t-1}	a_2^{t-1}	Index
1	1	1	1	0
1	1	1	0	1
1	1	0	1	2
1	1	0	0	3
1	0	1	1	4
1	0	1	0	5
1	0	0	1	6
1	0	0	0	7
0	1	1	1	8
0	1	1	0	9
0	1	0	1	10
0	1	0	0	11
0	0	1	1	12
0	0	1	0	13
0	0	0	1	14
0	0	0	0	15

Table 2: **Relationship between interaction state and index.**

under the definition of MTBR. Such a distribution remains a mathematically admissible class, but it is unattainable along the realized path of play and thus irrelevant to determining whether MTBR is a partner strategy.

Accordingly, MTBR qualifies as a partner strategy whenever the benefit-to-cost ratio $(b/c)^*$ exceeds 1.5, as we have previously shown.

Comments:

To my understanding, the reason why MTBR appears successful in their experiments is due to the following factors. Firstly, they eliminated execution error entirely while assuming that the initial move is randomly chosen, which is a rather specific assumption. (While Fig.R1 demonstrates their study of the case with noise, those instances are relatively rare since they only continue the game for 20 rounds.) Secondly, most of the strategies they considered maintain cooperation if the initial moves in the first few rounds were successful. MTBR is designed to excel in the initial rounds with those specific opponent strategies. Additionally, they primarily focus on repeated games with a length of 20 rounds, which strikes a balance between being neither too short nor too long. If the length is shorter, defecting strategies are more likely to be chosen. Conversely, if the length is longer (such as 1000 or 10000), the advantage of the initial moves diminishes, and they would exhibit similar performance to other strategies.

We thank the referee for raising the concern about interaction length. In response, we performed additional experiments extending the interaction length to 1000 rounds under noise-free conditions. The results (Fig. R2) show that MTBR remains highly successful: in about 90% of independent runs it ultimately takes over the entire population, while GTFT0.3 dominates in the remaining 10% of runs. This confirms that MTBR’s cooperative advantage is not restricted to the 20-round setting. We note that when the interaction length is very long (or in the limit of infinitely repeated games), the self-play payoff of most cooperative strategies converges to R , thereby reducing performance differences among them. Nevertheless, MTBR maintains a relative advantage, though this advantage becomes proportionally diluted as the horizon increases.

At the same time, We emphasize that the present study focuses on noise-free environments, where the search for dominant strategies provides novel insights into the emergence of cooperation. Systematically examining noisy environments is an important extension that requires substantial additional work, and we plan to address this in a follow-up study. We also acknowledge that execution errors gradually weaken the advantage of MTBR, as is generally the case for many cooperation-promoting strategies, and we have clarified this scope of applicability in the revised manuscript.

Finally, we emphasize that when the horizon is extremely short, even highly effective cooperative strategies cannot calibrate behavior or establish sustained reciprocity. In such cases, many strategies, including MTBR, fail to succeed. Indeed, one of the central purposes of studying repeated games is to understand how extended interactions stabilize and enhance cooperation.

We have incorporated these points into the Discussion section.

Fig. R2: **Evolutionary dynamics with long interactions.** We consider an evolving population based on strategy set 2 from the main text. The interaction length is fixed at 1,000 rounds. Initially, all strategies are equally abundant. Each line shows the average frequency across 1,000 independent runs.

Replies to comments and suggestions from referee 3

Comments:

I appreciate the authors' revisions and efforts. All my concerns have been addressed, I have no further comments and recommend acceptance.

We thank Reviewer #3 for their careful reading and constructive feedback in the previous round, which greatly helped us improve the manuscript. We are glad that all of the reviewer's concerns have now been addressed and that they recommend acceptance.

Replies to comments and suggestions from referee 4

Comments:

The revised MS is good enough to persuade me evaluating that this version can be acceptable to the journal. It sincerely responds all the points and suggestions the reviewers gave. I'm glad to draw a positive evaluation on the MS.

We appreciate Reviewer #4's thorough assessment and positive comments on our revision. We are encouraged by the reviewer's recognition that we have carefully addressed all the earlier points and suggestions.

Replies to comments from referee 2

Comments:

The authors have addressed many of the concerns raised in my previous review and have improved the manuscript by reporting the limitations of the proposed strategy in more objective ways and providing additional analysis. I appreciate the clarification regarding my misunderstanding of the definition of partner strategies, and I understand that the paper only considers the error-free case.

Thank you for your positive feedback and for your understanding of the main focus of our manuscript.

As the authors mentioned, they consider only the error-free case. (Although they present some results with noise, the main focus is on the error-free case.) Why then does canonical TFT or GRIM not perform well in this setting? As I pointed out in my previous review, the success of MTBR (unsuccessfulness of TFT and GRIM) is likely due to the specific choice of the initial condition. While the authors are focusing on the error-free case, they assume random initial moves, which seems quite inconsistent. If we consider the error-free and finite-rounds games, it is natural to assume that both players start with action prescribed by the strategy. In that case, I suspect that even canonical TFT or GRIM would perform as well as MTBR. As shown in Table 1, the prescription of MTBR is almost identical to that of TFT, with the only difference being in the actions for the mutual defections in the last two rounds. MTBR's apparent success appears to hinge on the assumption that the initial moves are random. I don't think MTBR is particularly superior to TFT as their moves are almost identical.

In the revised manuscript, we have stressed that our study actually examines both the error-free and error-based cases. Specifically, we provide a theoretical analysis of the evolutionary dynamics (see Supplementary Note 2) and proofs of partner strategies (see Supplementary Note 3) in the error-free case, and we demonstrate the robustness of MTBR through simulations in the error-based case (see Lines 243-258). In both settings, the initial actions are randomized, creating a more challenging environment for the emergence of cooperation, particularly for strategies such as TFT and GRIM. The superior performance of MTBR over classical strategies highlights its enhanced ability to recover efficiently from random initial conditions, ensuring consistently strong performance even under stochastic circumstances.

Furthermore, MTBR fundamentally differs from TFT: whereas TFT relies solely on the opponent's previous move, MTBR considers at least 20 distinct states when determining its actions (see Lines 118-144, Supplementary Note 1). This expanded state awareness enables MTBR to adapt more flexibly and robustly to diverse game dynamics, thereby offering a clear advantage over TFT in our analysis.

The amount of improvement that MTBR causes (Fig. 4c, 4f, and also seen in Fig. 3) is not

significant. As I pointed out in my first comment and they replied and revised in their second submission, the amount of the improvement is very small (2.9 - $\dot{}$ 2.938) although it visually looks significant in their plots, (which I find rather misleading.) They defended their argument claiming that the theoretical maximum is 3.0 and the improvement is 38% of the theoretical limit. However, this is a nonsense argument. If the improvement were 2.98 - $\dot{}$ 2.99, the improvement would be 50% of the theoretical limit and even more significant according to their argument. Of course, such an improvement would be insignificant.

In the revised manuscript, we have further addressed the significance of the improvement achieved by MTBR (see Lines 217-232).

First, GTFT is widely recognized as one of the most successful strategies for promoting high levels of cooperation within populations. When GTFT dominates, it drives the population to an average payoff of 2.90, approaching the theoretical maximum of 3.00. Remarkably, MTBR is able to further enhance cooperation, yielding an even higher average payoff of 2.93. Although this numerical improvement appears marginal, it is nontrivial given the narrow range between 2.90 and the theoretical limit. More importantly, its impact on the evolutionary dynamics is substantial. The introduction of MTBR transforms a mixed population, previously composed of multiple coexisting strategies, into a near-monoculture dominated by MTBR, leading to a qualitatively distinct evolutionary outcome.

Second, the cooperation-promoting effect of MTBR becomes even more evident in scenarios where GTFT fails to maintain dominance. For example, in a small population of approximately 50 individuals, the absence of GTFT0.3 results in a low average payoff of 1.78. Introducing GTFT0.3 raises the average payoff to 2.44, indicating partial cooperation. In contrast, introducing MTBR instead leads to a much higher average payoff of 2.94, nearly achieving full cooperation. This demonstrates that MTBR not only surpasses GTFT0.3 in challenging evolutionary settings but also drives populations toward near-complete cooperation. Furthermore, when both MTBR and GTFT0.3 are present, the resulting dynamics again produce a high average payoff, underscoring MTBR's robustness and its strong capacity to promote cooperation across different conditions.

The dominance of the MTBR in evolutionary settings may be significantly enhanced in their evolutionary simulations (Fig. 4b). As we see in Fig.3, the payoffs of MTBR and some of the other strategies (such as GTFT) are similar. Nevertheless, in Fig.4b, MTBR looks outstanding. Once the other aggressive strategies are wiped out and a cooperative environment is established, what matters is the self-cooperation levels. Even if the advantage of MTBR's self-cooperation level is minor (according to Supplementary Figure 2, MTBR's self-payoff is 2.94 while that for Gradual TFT is 2.91), such an advantage may look significantly enhanced in their settings, possibly due to large N.

The remarkable dominance of MTBR in evolutionary settings further reinforces its significance, as discussed in the previous comment. In the revised manuscript, we have addressed that this dominance persists across a broad range of population sizes, remaining robust even when the

population size is as small as $N=150$ (see lines 452-454).